# The rare sugar D-tagatose protects plants from downy mildews and is a safe fungicidal agrochemical

Susumu Mochizuki [1,3], Takeshi Fukumoto[1,2,3], Toshiaki Ohara[2,3], Kouhei Ohtani[1], Akihide Yoshihara[1], Yoshio Shigematsu [2], Keiji Tanaka[2], Koichi Ebihara[2], Shigeyuki Tajima[1], Kenji Gomi[1], Kazuya Ichimura [1], Ken Izumori[1] & Kazuya Akimitsu [1✉]

The rare sugar D-tagatose is a safe natural product used as a commercial food ingredient. Here, we show that D-tagatose controls a wide range of plant diseases and focus on downy mildews to analyze its mode of action. It likely acts directly on the pathogen, rather than as a plant defense activator. Synthesis of mannan and related products of D-mannose metabolism are essential for development of fungi and oomycetes; D-tagatose inhibits the first step of mannose metabolism, the phosphorylation of D-fructose to D-fructose 6-phosphate by fructokinase, and also produces D-tagatose 6-phosphate. D-Tagatose 6-phosphate sequentially inhibits phosphomannose isomerase, causing a reduction in D-glucose 6-phosphate and D-fructose 6-phosphate, common substrates for glycolysis, and in D-mannose 6-phosphate, needed to synthesize mannan and related products. These chain-inhibitory effects on metabolic steps are significant enough to block initial infection and structural development needed for reproduction such as conidiophore and conidiospore formation of downy mildew.

[1] International Institute of Rare Sugar Research and Education & Faculty of Agriculture, Kagawa University, 2393, Miki, Kagawa 761-0795, Japan. [2] Agrochemical Research Center, Mitsui Chemicals Agro, Inc., 1358 Ichimiyake, Yasu, Shiga 520-2362, Japan. [3]These authors contributed equally: Susumu Mochizuki, Takeshi Fukumoto, Toshiaki Ohara. ✉email: akimitsu.kazuya@kagawa-u.ac.jp

Rare sugars are defined by the International Society of Rare Sugars as monosaccharides that are rarely present in nature[1,2]. Saccharides are produced by photosynthesis, and most are stored as polysaccharides in the form of starch, comprising monomers of D-glucose, the most abundant hexose monosaccharide on earth. Among the 34 hexoses are 16 aldoses, 8 ketoses, and 10 polyols; four hexoses (D-glucose, D-fructose, D-mannose, and D-galactose) are profuse in nature, while the other 30 hexoses are rare sugars (Supplementary Figs. 1, 2)[1,2]. In 2002, Izumori presented the Izumoring concept (Supplementary Fig. 2)[2,3], a blueprint for the enzymatic synthesis of all hexoses including the 30 rare ones.

D-Allulose (previously called D-psicose or systematically called D-ribo-2-hexulose) (Supplementary Fig. 1) is the most well-known rare sugar. D-Allulose is an epimer at the carbon 3 position of D-fructose (Supplementary Fig. 1) and has antihyperlipidemic and antihyperglycemic effects that decrease adipose tissue mass in animals and humans, providing a potential use as a zero-calorie functional sweetener[4–10]. D-Allulose and D-allose, a C-3 epimer of D-glucose (Supplementary Fig. 1), also have an anti-aging effect as shown through lifespan studies of the nematode *Caenorhabditis elegans*, a model organism for longevity research[11,12]. For plants, D-allulose and D-allose can induce disease tolerance and/or growth inhibitory effects in *Oryza sativa* (rice), *Arabidopsis thaliana*, and *Lactuca sativa* (lettuce)[13–19]. The growth inhibition by D-allose is induced by a regulatory mechanism related to the plant growth hormone gibberellin, and the regulatory effects are driven through transduction of a gibberellin signal rather than gibberellin biosynthesis itself[13,15]. The disease-tolerance effect shown in rare sugar-treated plants is induced by a transient generation of reactive oxygen species following induction of typical plant defense responses such as pathogenesis-related protein (PR-protein) gene expression and lesion mimic formation similar to the hypersensitive reaction in plants or apoptosis in animals[13–17]. Interestingly, phosphorylation at the C-6 position of these rare sugars in plant cells is essential for these effects;[13,15,16] thus, various hexose kinases are considered as the target enzyme for analyzing the modes of action and signal transduction of these rare sugars[13,15,16].

Because the Izumoring concept[2,3] (Supplementary Fig. 2) made enzymatic production of all hexosaccharides possible, rare hexose sugars in addition to D-allulose and D-allose are also available now to test their disease control efficacy as agrochemicals. Here, we focused on evaluating D-tagatose, a C-4 epimer of D-fructose (Supplementary Fig. 1), to control various diseases and examined its mode of action using *A. thaliana* and *Hyaloperonospora arabidopsidis* isolate Noco2 as a host-pathogen model system. D-Tagatose was found to be produced naturally when milk is overheated during conventional sterilization[20], and it was declared as safe by the Food and Drug Administration (FDA) in the United States and the World Health Organization (WHO) in 2001[21,22]. D-Tagatose has been produced industrially using various methods such as mixing D-galactose with calcium hydroxide under alkaline conditions or enzymatic isomerization[23]. It is also commercially available as a food and beverage additive and sweetener. We here describe its unique mode of action, inhibiting the metabolism of mannose as an energy source in phytopathogenic oomycetes, and its potential as an innovative, eco-friendly agrochemical.

## Results

**D-Tagatose effects on plant diseases.** Effective concentrations (w/v) of D-tagatose that were expected to reduce severity to 50% that of the mock treatment (untreated) were examined in either pot or field trials or both against multiple diseases (Table 1). D-Tagatose solution at 0.5–10% (w/v) reduced the severity of symptoms induced by all pathogens tested, and even at 0.5% or 1% (w/v) of D-tagatose significantly reduced the severity of downy mildew and powdery mildew on a wide range of host plants (Table 1). Typical symptoms of cucumber downy mildew and the reduced symptoms by these treatments with D-tagatose are shown in Fig. 1a. These treatments against cucumber downy mildew indicated that the pathogen-inoculated first true leaf after treatment with 5% (w/v) D-tagatose showed no symptoms, the same as the effect of treatment with agrochemicals such as probenazole [PBZ], acibenzolar-S-methyl [ASM], or metalaxyl (Fig. 1a). No phytotoxicity was observed on plants treated with D-tagatose, while development of the second true leaf was significantly inhibited on plants treated with D-allose or D-allulose (Fig. 1a).

Disease severities on cucumber leaves treated with 1 or 5% D-tagatose or metalaxyl at different timings before or after inoculation with the downy mildew pathogen were compared to verify whether D-tagatose works as a preventive or curative agent (Fig. 1b). The inhibition of disease development by D-tagatose treatment was similar to the inhibition caused by fungicide treatment; no symptoms were observed when either the sugar or fungicide was applied 5 days before inoculation, and 90% or more of symptoms were suppressed when these agents were applied even at 7 days before the inoculation (Fig. 1b, Preventive). Moreover, no symptoms were observed after either of these treatments was applied within 48 h after inoculation, but D-tagatose treatment at 60 h or later after inoculation and fungicide treatment at 72 h or later after inoculation did not inhibit symptom development completely (Fig. 1b, Curative).

In field trials, grapevine, cucumber, Chinese cabbage, onion, and spinach were then treated with or without 1 or 5% (w/v) D-tagatose, fungicides (cyazofamid FL [CZF], chlorothalonil [CTN] FL, and mancozeb ([MCZ] WP) with inoculation of one of five downy mildew pathogens (detail experimental conditions were shown in Supplementary Table 2). In all cases, the suppression of symptoms by D-tagatose was similar to that by the fungicides (Fig. 1c–g).

**Effects of D-tagatose on Arabidopsis downy mildew.** In a test of a host-pathogen model system, symptoms on *A. thaliana* ecotype Colombia-0 (Col-0) caused by *H. arabidopsidis* isolate Noco2 were suppressed by 1% (w/v) D-tagatose (Fig. 2a), indicating its efficacy in this host-pathogen system. To understand the mode of action of D-tagatose using the model system, we assessed the extent of hyphal growth of isolate Noco2 on cotyledons of *A. thaliana* treated with different concentrations of the rare sugar (Fig. 2b, c). Microscopic observations of hyphal growth of *H. arabidopsidis* isolate Noco2 at 10 days after the inoculation of Arabidopsis seedlings indicated growth was inhibited in leaves treated with 1% (w/v) D-tagatose (Fig. 2b). At 7 days after inoculation, hyphal growth was rated based on hyphal growth efficiency (% of leaf area with hyphae). The percentage of leaves with extensive hyphal growth (+++ rating) decreased, with a negative correlation to the D-tagatose concentration, and the percentage of leaves with no fungal growth (– rating) increased, with a positive correlation to concentration, with a dose-dependency in both associations (Fig. 2c). After treatment with 2.5 mM D-tagatose, 58.0 ± 1.8% of the cotyledons had 75–100% area (+++ rating) of their leaves with hyphae similar to that of mock-treated (no-sugar treatment) leaves (57.5 ± 1.8%) (Fig. 2c). Treatment with 25–100 mM D-tagatose lowered the percentage of cotyledons with 75–100% colonization to 9.9 ± 3.1% to 6.7 ± 0.3%, respectively (Fig. 2c).

When the effects of D-tagatose on asexual reproduction were examined, even 2.5 mM of D-tagatose significantly inhibited conidiation on seedlings with four leaves (including the

**Table 1 Effective dose of D-tagatose on various diseases caused by different pathogens on different host plants in pot and field trials.**

| Class | Disease | Pathogen | Host | Effecive Dose (%)[a] | | Reference |
|---|---|---|---|---|---|---|
| | | | | Pot trials | Field trials | |
| Oomycetes | Downy mildew | *Plasmopara viticola* | Grapevine | 1 | 3 | 44,50 |
| | | *Pseudoperonospora cubensis* | Cucumber | 0.5 | 0.5 | 44,50, this study |
| | | *Hyaloperonospora parasitica* (syn. *H. brassicae*) | Chinese cabbage | nt. | 1 | This study |
| | | *Hyaloperonospora parasitica* (syn. *H. brassicae*) | Cabbege | 1 | nt. | 44,50 |
| | | *Peronospora destructor* | Onion | nt. | 1 | This study |
| | | *Peronospora farinosa* f. sp. *spinaciae* | Spinach | nt. | 1 | This study |
| | Damping off | *Pythium aphanidermatum* | Cucumber | 1 | nt. | 44 |
| | Seedling blight | *Pythium graminicola* | Rice | 1 | 1 | 44 |
| | Late blight | *Phytophthora infestans* | Tomato | 10 | nt. | 44 |
| | | *Phytophthora infestans* | Potato | nt. | 5 | This study |
| Ascomycetes | Powdery mildew | *Erysiphe necator* | Grapevine | 0.5 | nt. | 50 |
| | | *Podosphaera xanthii* | Cucumber | 0.5 | 0.5 | 44,50 |
| | | *Podosphaera leucotricha* | Apple | 0.5 | nt. | 50 |
| | | *Podosphaera aphanis* | Strawberry | nt. | 1 | This study |
| | | *Sphaerotheca fuliginea* | Eggplant | 0.5 | 0.5 | 50 |
| | | *Oidium violae* | Tomato | 0.5 | nt. | 50 |
| | | *Oidiopsis sicula* | Pepper | 0.5 | 1 | 50 |
| | | *Blumeria graminis* f. sp. *hordei* | Barley | 1 | nt. | 50 |
| | Gray mold | *Botrytis cinerea* | Tomato | >5 | nt. | 44 |
| | Alternaria sooty spot | *Alternaria brassicicola* | Cabbege | >5 | nt. | This study |
| | Brown spot | *Cochliobolus miyabeanus* | Rice | >5 | nt. | This study |
| | Anthracnose | *Colletotrichum orbiculare* | Cucumber | >5 | nt. | This study |
| | Blast | *Pyricularia oryzae* (syn. *Magnaporthe oryzae*) | Rice | 5 | nt. | 44 |
| Basidiomycetes | Brown rust | *Puccinia recondita* | Wheat | 5 | 5 | 44 |
| | Sheath blight | *Rhizoctonia solani* AG-1 IA | Rice | >5 | nt. | This study |

Experiments are described in Supplementary Tables 1 and 2.
*nt.* not tested.
[a]Effective dose is concentration (w/v) of D-tagatose that provides an expected disease control value of more than 50% against disease after mock treatment (untreated) based on disease severity.

cotyledons) ($p = 5.29e-9$), and more than 25 mM D-tagatose almost completely inhibited conidiation (Fig. 2d). D-Tagatose treatment similarly inhibited conidiophore and oospore formation on/in cotyledons (Fig. 2e, f), and formation of conidiophores and oospores (Fig. 2g) was suppressed almost completely by treatment with more than 25 mM D-tagatose. D-Tagatose deoxygenated at C-6 position (50 mM; hereafter "deoxyed") (Supplementary Fig. S1), however, did not inhibit conidiospore formation in planta (Fig. 2h). Hyphal length after conidiospores had germinated for 6 h in vitro on medium in microtiter plates was also suppressed by treatment with more than 25 mM D-tagatose, but no further dose response was observed even up to 400 mM (Fig. 2i, j). Treatment with the same concentrations of D-mannitol used as the control to check for a potential osmotic pressure effect did not show any effect even at 400 mM (Fig. 2j). The deoxyed D-tagatose (50 mM) (Supplementary Fig. 1) was also ineffective against in vitro conidiospore germination, whereas 50 mM D-tagatose reduced germination by 47.6% over that of the control (Fig. 2k). We did not find any morphological changes in shape or size of conidiospores, conidiophores or oospores by the treatment with any concentrations of D-tagatose (Fig. 2).

**D-Tagatose acts on pathogen but not host defense activation.** Toward understanding the mechanism underlying the D-tagatose effects on downy mildew, we analyzed the expression of PR-protein and defense-related genes: peroxidase [abbreviated here as POX], lipoxygenase [LOX], pore-forming toxin-like protein [Hrf]) and induced defense (4-coumarate-CoA ligase [4CL] and caffeoyl-CoA *O*-methyltransferase [CCoAMT]) in plants after

treatment with D-tagatose at the effective dose of 0.5% (w/v), which suppressed disease severity by more than 50% compared with the mock treatment in both pot and field trails (Table 1) and at 1% (w/v), which gave the same control as the fungicide treatment in a field test against cucumber downy mildew (Figs. 1d and 3a)[24–28]. Treatment with D-tagatose alone at 1% (w/v) did not significantly induce any of these genes within 48 h (Fig. 3a) or in susceptible cucumber leaves after inoculation with the pathogen without D-tagatose treatment. For all other combinations, gene expression did not change significantly within 48 h, except for minor inductions of LOX at 24 h after inoculation with D-tagatose treatment (Fig. 3a). When the expression of these genes was compared between inoculations with/without D-taga-tose treatment, only 4CL was induced at 24 h after inoculation in conjunction with D-tagatose treatment (Fig. 3a).

Since D-allulose and D-allose induced disease tolerance in rice as a plant activator by inducing PR-protein gene expression, which is typically initiated by a transient generation of reactive oxygen species after a hypersensitive reaction or apoptosis-like lesion mimic formation[14,16,17], we tested rice for an effect of D-tagatose on the PR-protein genes that were induced by D-allulose and D-allose. Overall expression patterns in scatter plots of microarray data at 2 days after treatment with 0.5 mM of D-allose[14], D-allulose[16,17], D-glucose[14,16,17], or D-tagatose indi-cated less fluctuation between the induction and reduction of the expression of all genes analyzed after D-tagatose treatment compared with the D-allose[14], D-allulose[16,17], or even D-glucose treatments[14,16,17] (Supplementary Fig. 3). Several PR-protein genes that are induced by D-allulose or D-allose in rice were not

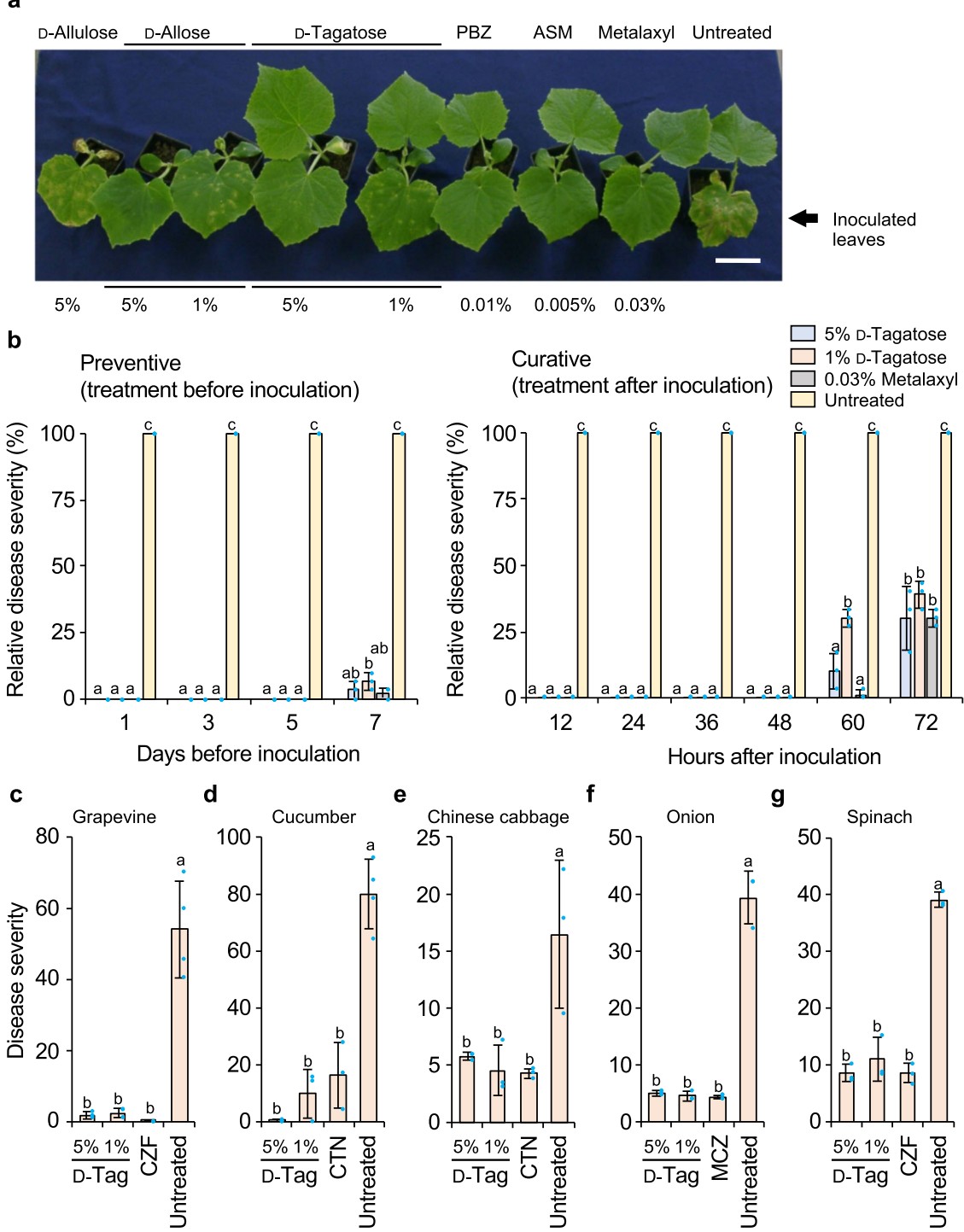

**Fig. 1 Effect of D-tagatose on severity of various downy mildews.** Typical symptoms of cucumber downy mildew on the first true leaf of untreated cucumber plant and reduced severity after treatment with 1% or 5% (w/v) D-tagatose, D-allulose, or D-allose in pot trials in comparison with probenazole [PBZ], acibenzolar-S-methyl [ASM], and metalaxyl (Scale Bar = 5 cm) (**a**). The timing of treatments with sugars and agrochemicals in pot trials varied from 1 to 7 days before inoculation with the pathogen ("Preventive") or from 12 to 72 h after inoculation ("Curative"), and their symptoms were compared using the averages of each severity after 7 days to calculate a relative disease severity against average severity for the mock-treated plants. Data presented are representative of three independent experiments (**b**). Severity of downy mildews in field trials after treatment with D-tagatose (1% or 5%, w/v) (D-Tag) compared (n = 50 to 163 leaves per group) with a fungicide: cyazofamid FL (CZF) at 94 parts per million (ppm) on grapevine and spinach downy mildews (**c**, **g**), chlorothalonil (CTN) FL at 400 ppm for cucumber and Chinese cabbage downy mildews (**d**, **e**), and mancozeb (MCZ) WP at 1875 ppm for onion downy mildew (**f**), or mock treatment (Untreated) (**c**–**g**). Disease severity was calculated from the degree of disease index based on the Fungicide Evaluation Manual by the Japan Plant Protection Association for field trials[51] as described in the Methods. Data presented are representative of 3–4 independent experiments. Error bars are SD. Means with different letters differed significantly at p < 0.05 in a Tukey–Kramer multiple comparison test.

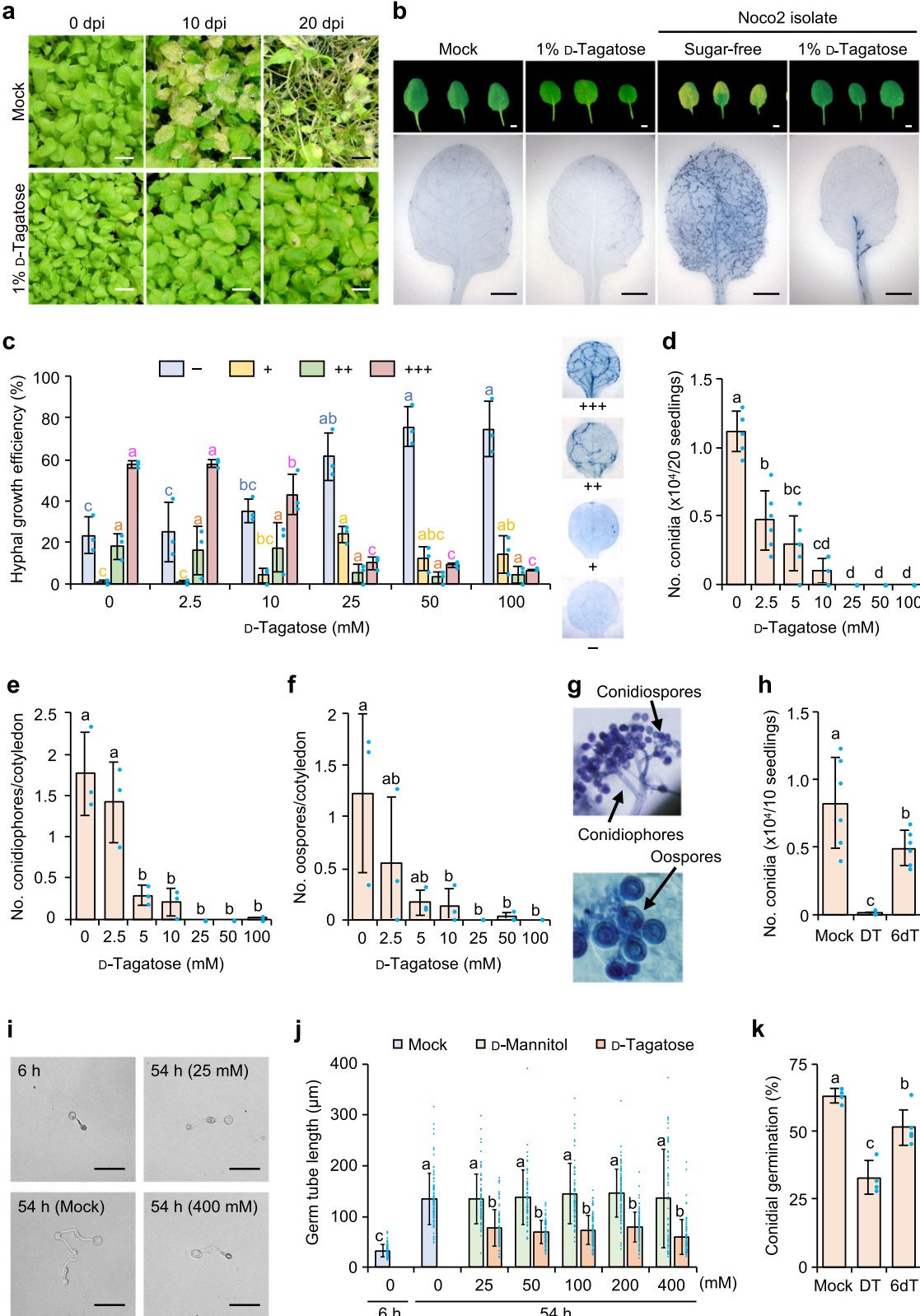

induced significantly by D-tagatose (Fig. 3b), while the expression of all these genes was stimulated significantly by D-allulose or D-allose treatment as described previously[14,16,17].

Moreover, RNA-sequencing did not show any major up- or downregulation in the expression of PR-proteins, plant hormones (e.g., jasmonic acid, ethylene, and salicylic acid)-related and defense-related genes between D-tagatose-treated and untreated plants of *A. thaliana* after inoculation with *H. arabidopsidis* isolate Noco2 (Fig. 3c, Supplementary Fig. 4a). Some genes were up- or downregulated more than 10 times between the inoculated plants that were treated with D-tagatose or untreated (Supplementary Fig. 4b), but no particular trend for any group of genes

**Fig. 2 Effects of D-tagatose on a plant host–oomycete pathogen model system, *Arabidopsis thaliana* ecotype Col-O–*Hyaloperonospora arabidopsidis* isolate Noco2.** Disease severity and pathogen development after treatment of leaves with D-tagatose at 0–400 mM or 1% (w/v) (ca. 50 mM), water (Control) or other sugars. Symptoms at 0, 10, or 20 days post inoculation (dpi) (Scale bar = 1 cm) (**a**). Typical hyphal growth in inoculated leaves with or without 1% D-tagatose treatment (Scale bar = 2 mm) (**b**). Mean (±SD) hyphal growth on 111 to 138 cotyledons for each treatment at 7 days after inoculation was rated based on percentage of entire leaf area with hyphae: +++ for 75 to 100% of leaf; ++ for 25% to 74%; + for <24%; and – for no hyphae (Three independent replications were done); typical hyphal growth for each rating is shown in the same panel (**c**). In planta production of conidiospores (**d**), conidiophores (**e**), and oospores (**f**). Data presented are representative of six (**d**) or three (**e**, **f**) independent experiments. Typical morphology of structures examined in panels **d–f** (images are 200 × 200 μm) (**g**). Effects of deoxygenated D-tagatose at position C-6 (6dT) (50 mM) compared with those caused by D-tagatose (DT) (50 mM) on conidial production in planta. Data shown are representative of six independent experiments (**h**). Typical conidial germination after 48 h on germination medium in microtiter plates with or without 25 or 400 mM D-tagatose treatment of conidia, which were pregerminated for 6 h without sugars (scale bar = 50 μm) (**i**). Length of germ tubes after 48 h on germination medium in microtiter plates with 0–400 mM D-tagatose or D-mannitol compared with conidia pregerminated for 6 h without these sugars. Representative of 60 to 93 independent germ tubes lengths were measured (**j**). Mean of conidial germination (±SD) after 24 h on germination medium in microtiter plates compared with germination on medium with 50 mM of D-tagatose (DT) or 6dT. Data presented are representative of 4–6 independent experiments ($n$ = 85–263 per group) (**k**). Means with different letters differed significantly at $p < 0.05$ in a Tukey–Kramer multiple comparison test.

was found to explain the protection conferred by D-tagatose against downy mildew. Several sugar-related genes such as *At5g45830* (*DELAY OF GERMINATION 1* [*DOG1*]), *At5g40900* (glycosyltransferase), *At2g33100* (cellulose synthase-like protein [*ATCSLD1*]), and *At5g50800* (sucrose efflux transporter [*SWEET13*]) were among the genes that changed more than 10 times compared with those in the untreated *A. thaliana*, but the differences based on $q$-value were not significant in significance level of 0.05 nor is a direct relation to plant protection apparent for these genes (Supplementary Fig. 4b).

**Effects of D-tagatose on mannose metabolism in the pathogen.** Because conidial germination and germ tube lengths of isolate Noco2 were inhibited by direct treatment of the pathogen with D-tagatose in vitro (Fig. 2i–k), we evaluated the effects of D-tagatose on sugar metabolism in isolate Noco2 in vivo and in vitro. The first enzyme in the sugar metabolic pathway is a monosaccharide kinase that phosphorylates the C-6 position of hexoses, and the target peak indicating phosphorylation of D-tagatose at C-6 (for D-tagatose 6-phosphate) was detected from a reaction mixture of D-tagatose with crude enzyme extracts from conidiospores of the pathogen (Fig. 4a).

Rarely has an enzyme been found to be specific for a rare sugar as the major substrate in nature[1,2], but some enzymes involved in sugar metabolism do have a broad substrate specificity that includes rare sugars. Five genes for putative enzymes with a functional annotation for possible monosaccharide phosphorylation—fructokinase (LC500344 equivalent to Hpa812304 of *H. arabidopsidis* isolate Emoy2), glucokinase (LC500564 equivalent to Hpa800730), xylulose kinase (LC500562 equivalent to Hpa801075), ribokinase (LC500561 equivalent to Hpa805763), and galactokinase (equivalent to Hpa809752)—were selected from *H. arabidopsidis* isolate Noco2 based on information from the EnsemblProtists database (http://protists.ensembl.org/index.html) and the genomic sequence data for *H. arabidopsidis* isolate Emoy2 (NCBI: txid559515)[29]. Heterologous expression products from four of the five putative enzyme-encoding genes in the Noco2 isolate responsible for D-tagatose phosphorylation were expressed in *Escherichia coli* to examine potential phosphorylation of D-tagatose at C-6 (Supplementary Fig. 5). Among these genes, the gene encoding galactokinase was not expressed well in the *E. coli* system and could not be tested for D-tagatose phosphorylation. GST-tagged expression product (58 kDa) from the gene encoding fructokinase (LC500344) had significant kinase activity (Fig. 4b, c), while the other three enzymes did not (Supplementary Fig. 5). Fructokinase (LC500344) phosphorylated both D-fructose and D-tagatose to generate, respectively, D-fructose 6-phosphate and D-tagatose 6-phosphate, with $K_m$ = 0.197 ± 0.044 mM, $V_{max}$ = 3.48 ± 0.89 μmol mg protein$^{-1}$ min$^{-1}$

using D-fructose, and $K_m$ = 52.67 ± 22.75 mM, $V_{max}$ = 0.15 ± 0.04 μmol mg protein$^{-1}$ min$^{-1}$ using D-tagatose (Supplementary Fig. 6a–d). The optimal temperature for enzyme activity was 35 °C, and this enzyme was stable at lower than 40 °C (Supplementary Fig. 6e, f). The enzymatic activity was optimal at pH 10.0 to 11.0 and stable above pH 6, indicating that this enzyme was relatively stable in basic conditions (Supplementary Fig. 6g, h). Enzymatic activity was lost after EDTA treatment to chelate ions, and the highest activity was recovered after the addition of $Co^{2+}$, $Fe^{2+}$, or $Mg^{2+}$ (Supplementary Fig. 6i), respectively.

Since both D-fructose and D-tagatose served as a substrate for the same fructokinase, we tested them together for competitive inhibition of the other and analyzed D-fructose kinase activity by monitoring D-fructose 6-phosphate production. With a $K_i$ for D-fructose of 70.05 ± 5.14 mM, D-tagatose produced less D-fructose 6-phosphate (Fig. 4d and Supplementary Fig. 6j). On the basis of Dixon plot analyses (Fig. 4d), addition of 250 mM D-tagatose was calculated to inhibit the production of D-fructose 6-phosphate by a maximum of 41.0% when combined with 0.5 mM D-fructose and by 13.1% when 50 mM D-tagatose was combined with 0.5 mM D-fructose (Supplementary Fig. 6k).

Because of the competitive inhibitory effect of D-tagatose on fructokinase, we also examined D-tagatose treatment on the next enzyme in D-fructose metabolism; D-fructose is next phosphorylated by fructokinase to D-fructose 6-phosphate, the main energy source for glycolysis. However, D-fructose 6-phosphate can also be converted by phosphomannose isomerase into D-mannose 6-phosphate, the first intermediate in the mannan synthetic pathway. In addition, phosphomannose isomerase can catalyze the interconversion between D-fructose 6-phosphate and D-mannose 6-phosphate, and interestingly, in the process, forms a *cis*-enediol intermediate, the same intermediate formed by phosphoglucose isomerase. Thus, this enzyme is often a bifunctional phosphomannose/phosphoglucose isomerase mediated by a *cis*-enediol intermediate (Fig. 5a).

To examine the effect of D-tagatose 6-phosphate on phosphoglucose isomerase, a gene encoding phosphomannose isomerase (LC500563 equivalent to Hpa802576) was selected from the EnsemblProtists database (http://protists.ensembl.org/index.html) based on the genomic sequence data of *H. arabidopsidis* isolate Emoy2 (NCBI: txid559515)[29], and heterologous expression products from this gene in isolate Noco2 were expressed in *E. coli* to assay enzymatic activities. As we expected, His-tagged expression products (45 kDa) from the gene encoding phosphomannose isomerase (LC500563) had significant dual activities, and both D-glucose 6-phosphate and D-mannose 6-phosphate were detected after the addition of D-fructose 6-phosphate (Fig. 5b, c). Because D-tagatose is converted to D-tagatose 6-

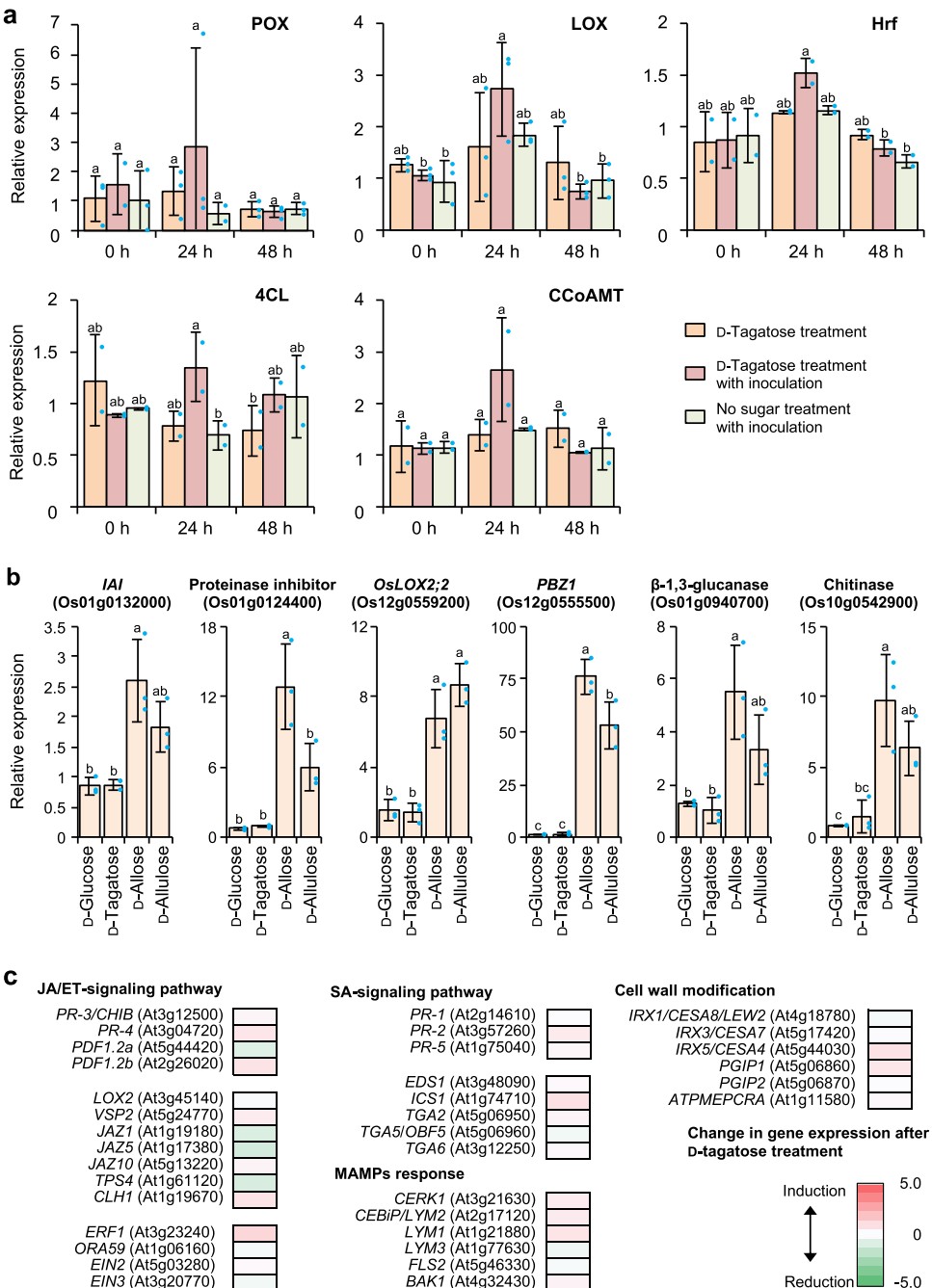

**Fig. 3 D-Tagatose effects on expression of PR-protein and defense-related genes in cucumber, rice, and Arabidopsis plants.** RT-qPCR analyses of expression of cucumber PR-protein and defense-related genes after cucumber treatment with D-tagatose with or without inoculation with *Pseudoperonospora cubensis*. Expression is relative to that in mock-treated leaves (mean ± SD of 2 or 3 independent replications; means were compared using a Tukey–Kramer multiple comparison test; different letters indicate a significant difference among means, *p* < 0.05). Transcript levels were normalized by comparison with actin (AB010922). Abbreviations are POX (peroxidase), LOX (lipoxygenase), Hrf (pore-forming toxin-like protein), 4CL (4-coumarate-CoA ligase) and CCoAMT (caffeoyl-CoA *O*-methyltransferase), respectively (**a**). Relative expression of rice PR-protein genes after rice treatment with various sugars against those without sugar treatment was calculated using data from microarray analyses (mean ± SD of three independent replications; means were compared using a Tukey–Kramer multiple comparison test; different letters indicate a significant difference among means, *p* < 0.05) (**b**). Color-coded differential gene expression based on RNA-seq analyses of *Arabidopsis thaliana* ecotype Col-0 with or without D-tagatose treatment and inoculation with *Hyaloperonospora arabidopsidis* isolate Noco2 (**c**). Statistical analyses based on *q*-values of the respective data in panel **c** are described in Supplementary Fig. 4a.

phosphate in *H. arabidopsidis* isolate Noco2 by fructokinase, which leads to the inhibition of phosphorylation of D-fructose to D-fructose 6-phosphate, we also examined the effect of D-tagatose 6-phosphate addition on the activity of phosphomannose isomerase; D-tagatose 6-phosphate (12.5–37.5 mM) significantly inhibited the production of D-mannose 6-phosphate by 51.6 (*p* = 8.85e-7) to 60.8% (*p* = 2.47e-7) and D-glucose 6-phosphate by 19.3 (*p* = 3.53e-4) to 38.2% (*p* = 2.25e-6) (Fig. 5d).

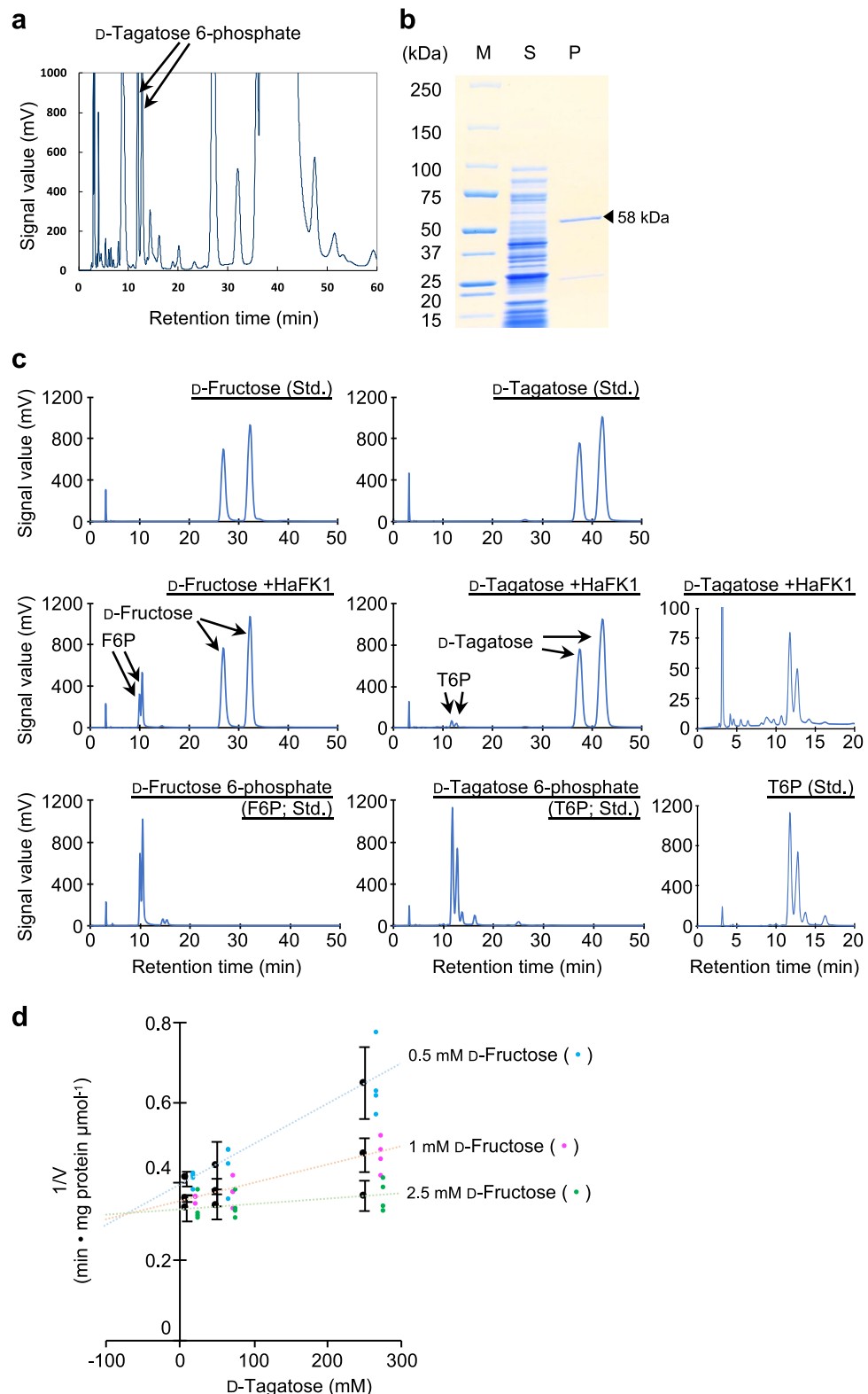

## Discussion

Control of crop diseases is essential for maintaining a sufficient yield of agriproducts for food production. Synthetic fungicides are generally used to efficiently control diseases, but natural eco-friendly products to protect crop against diseases are in demand.

In the present study, D-tagatose reduced severity of numerous plant diseases by at least 50% in pot or field trials; as little as 0.5%

(w/v) was required against downy mildew and powdery mildew of numerous economically important crops. The effectiveness of D-tagatose against the various diseases tended to differ, and it was more effective against biotrophs than necrotrophs or hemi-biotrophs, especially against downy mildews (oomycetes) and powdery mildew (ascomycetes). Interestingly, about 50 mM D-tagatose inhibited symptoms caused by several downy mildews

**Fig. 4 Competitive inhibition of D-fructose phosphorylation activity of fructokinase from *Hyaloperonospora arabidopsidis* isolate Noco2 by D-tagatose.** HPLC profile of reaction products containing crude enzyme extracts from conidiospores and D-tagatose contained D-tagatose phosphorylated at the C-6 position (D-tagatose 6-phosphate, arrows indicate corresponding peaks) (**a**). A candidate enzyme for the phosphorylation of D-tagatose was identified as a fructokinase (LC500344); the heterologous expression product of fructokinase has a molecular weight of 58 kDa in SDS-PAGE (**b**). S and P in panel **b** indicate soluble fraction (S) and purified protein (P), respectively. The phosphorylation activities of the enzyme for D-tagatose and D-fructose were determined by HPLC analyses (**c**). Spectra for authentic compounds in panel **c** are marked with "Std.". Dixon plot analysis of the kinetics of fructokinase for D-fructose (0.5–2.5 mM) at different concentrations of D-tagatose (10–250 mM) indicated competitive inhibition by D-tagatose with $1/V_{max} = 0.321$ ($V_{max} = 3.12$ μmol mg protein$^{-1}$ min$^{-1}$) with $K_i$ for D-fructose of 70.05 ± 5.14 mM (**d**). Data presented are representative of four independent experiments. Error bars are SD.

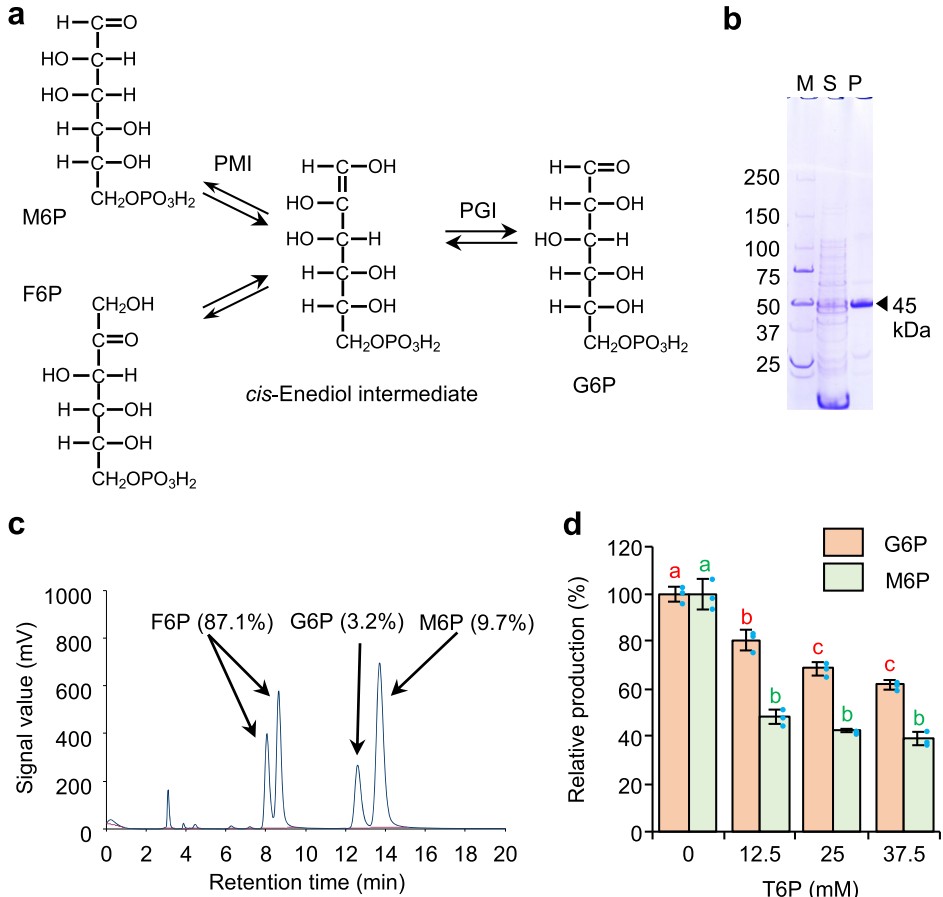

**Fig. 5 Competitive inhibition of phosphomannose isomerase from *Hyaloperonospora arabidopsidis* isolate Noco2 for D-fructose 6-phosphate catalysis by D-tagatose 6-phosphate treatment.** Phosphomannose isomerase (PMI) often has bifunctional phosphomannose/phosphoglucose isomerase (PGI) activity mediated by a *cis*-enediol intermediate (**a**). A candidate enzyme for *H. arabidopsidis* isolate Noco2 was identified as a phosphomannose isomerase (LC500563), and heterologous expression products of the phosphomannose isomerase with a molecular weight of 45 kDa in SDS-PAGE (**b**). S and P in panel **b** indicate soluble fraction (S) and purified protein (P), respectively. The enzyme had significant dual activity; both D-glucose 6-phosphate (G6P) and D-mannose 6-phospohate (M6P) were detected on HPLC analysis by adding D-fructose 6-phosphate (F6P) as the substrate (**c**). The dual activity of the enzyme for F6P was inhibited significantly by addition of D-tagatose 6-phosphate (T6P) (12.5–37.5 mM) for both M6P and G6P production (**d**). Data presented are representative of three independent experiments. Means (±SD) were compared using a Tukey–Kramer multiple comparison test; different letters above the bars indicate a significant difference ($p < 0.05$).

to a level similar to that provided by fungicide treatments. D-Tagatose was an effective disease control agent and did not cause any visible phytotoxicity such as leaf growth inhibition, which is caused by D-allose and D-allulose[13–17]. D-Tagatose had a 7-day residual effect, indicating that applications at 7-day intervals would be practical. Interestingly, D-tagatose also had some curative effect, and no symptoms were observed when D-tagatose was applied at 48 h after the inoculation as well. Although D-tagatose has recently received much attention as an ingredient and sweetener for various foods and beverages because of its sucrose-like taste, with 90–92% less sweetness and less than half

the calories (1.5 kcal g$^{-1}$) of sucrose, its antidiabetic and weight loss potential, association with beneficial increases in HDL, and possible use in managing diabetes/obesity[30,31], the fungicidal effects and a mechanism to explain the effects described here were not reported. Thus, we examined the mode of action of D-tagatose to understand the differences in the efficacy of D-tagatose against the different types of diseases.

Our examinations on the mode of action of D-tagatose, starting with a search of gene expression profiles for cucumber, rice and Arabidopsis, showed that it likely does not have a direct effect on plants as a plant defense activator; the expression patterns of

entire genes including several defense-related genes did not show any typical induction/or reduction after D-tagatose treatment. We also reported previously that D-tagatose treatment had no effect on the physiological function of tomato including sugar content, blossom end rot, water potential, stomatal conductance, or relative water content[32]. Toward more understanding of the defense mechanisms underlying the symptom reduction in D-tagatose-treated plants, we examined the effects of D-tagatose in detail using *A. thaliana* and its downy mildew pathogen, *H. arabidopsidis* isolate Noco2, as a model host-pathogen system. Treatment with D-tagatose significantly reduced symptom development in this model pathosystem, and concentrations of more than 25 mM of D-tagatose almost completely suppressed conidiophore and oospore formation. D-Tagatose at more than 25 mM dramatically suppressed symptoms, growth of hyphae, and formation of conidiospores, conidiophores and oospores without inducing any tested defense-related genes in the host plants, but direct treatment of the pathogen in vitro with D-tagatose at 50 mM inhibited germination by about 50% and subsequent germ tube elongation by 50%. Although the inhibitory effects on germination and germ tube elongation by D-tagatose were not as dramatic as we expected based on the level of inhibition of various developmental stages of this obligate parasite in its host tissues[33], conidiospores for in vitro experiments were always freshly collected after inoculation of their host *A. thaliana* and the condition/nutritional status of the conidiospores likely varied among the different harvests, which might account for differences in the effects of D-tagatose found in vitro and in planta.

Since D-tagatose directly affects the pathogen, we next examined potential mode of actions of the rare sugar on the downy mildew pathogen *H. arabidopsidis* isolate Noco2. As mentioned, the control by D-tagatose varied among the 25 plant diseases tested so far, and D-tagatose was most effective against downy mildews and powdery mildews. A typical feature of downy mildews is that their cell walls contain mannan[34-36], and based on a genome-scale metabolic network of the oomycete *Phytophthora infestans* and the derived context-specific models (CSMs) studies, fructose/mannose metabolism might play a pivotal role in the survival of oomycete pathogens in host plants[37]. Although D-mannose is a major component for saccharide chain synthesis in rats[38], D-mannose rarely enters the glycolytic system directly and is rarely metabolized in humans[39,40]. However, most organisms, likely including isolate Noco2, are generally considered to use hexokinase to convert D-mannose to D-mannose 6-phosphate, which is then converted by D-mannose 6-phosphate isomerase to D-fructose-6-phosphate for an energy source as a glycolytic intermediate[41]. D-Mannose is also used for mannan synthesis with D-mannose-6-phosphate as the first intermediate, but D-mannose in plants is known only in fruits and fruit peel at limited levels as a free monosaccharide, and D-mannose 6-phosphate is also often produced by the conversion of D-fructose-6-phosphate by D-mannose 6-phosphate isomerase[42]. The disruption of phosphomannose isomerase causes morphological abnormalities and loss of pathogenicity in *Cryotococcus neoformans* and inhibits hyphal growth in *Aspergillus nidulans*[41,42]. From these lines of evidence, mannose metabolism is thought to be important not only in oomycetes but also ascomycetes and basidiomycetes for energy metabolism and/or mannan synthesis. Thus, we examined the effect of D-tagatose on mannose metabolism, which synthesizes various types of mannan and related products in this study.

Fructokinase (LC500344) from *H. arabidopsidis* isolate Noco2 with a GST-tag (58 kDa) produced by a heterologous *E. coli* expression system had significant phosphorylation activity for both D-fructose and D-tagatose at the C-6 position to generate either D-fructose 6-phosphate or D-tagatose 6-phosphate (Fig. 4), respectively, while the other enzymes tested—glucokinase (LC500564),

xylulose kinase (LC500562), and ribokinase (LC500561)—did not phosphorylate D-tagatose (Supplementary Fig. 5).

Since both D-fructose and D-tagatose were shown to serve as substrates for the fructokinase, D-tagatose likely works as a competitive inhibitor of D-fructose kinase activity. Interestingly, D-tagatose 6-phosphate, the product from D-tagatose catalyzed by fructokinase, was also a possible competitive inhibitor of phosphomannose isomerase (LC500563), and addition of D-tagatose 6-phosphate at more than 37.5 mM significantly inhibited the production of about 60% of D-mannose 6-phosphate and about 38% of D-glucose 6-phosphate from D-fructose 6-phosphate (Fig. 5d). These results indicate that D-tagatose treatment (250 mM) inhibits about 41% of fructokinase phosphorylation of D-fructose (0.5 mM) to D-fructose 6-phosphate with production of D-tagatose 6-phosphate (Supplementary Fig. 6k), which also inhibits phosphomannose isomerase activities for conversion (1) between D-glucose 6-phosphate and D-fructose 6-phosphate for reducing glycolysis and (2) between D-fructose 6-phosphate and D-mannose 6-phosphate for reducing synthesis of mannan or mannoglucan, which are essential components of the walls of this pathogen[34-36].

The inhibitory effects caused by D-tagatose are not a simple block of fructokinase activity but rather a competitive inhibition at the maximum level, at its $K_i$, and the product (D-tagatose 6-phosphate, generated by fructokinase from D-tagatose) likely inhibits subsequent metabolism of phosphomannose isomerase in sequential reactions (Fig. 6). Because the inhibitory effect is not over the $K_i$ (about 70 mM of D-tagatose to fructokinase) and also is not a complete block of the enzymatic reaction, the lack of a dose-dependent inhibition by D-tagatose at 25 mM or higher of (1) hyphal growth in planta (Fig. 2c) or (2) germ tube elongation in vitro (Fig. 2j) are reasonable results. Concentrations of D-tagatose up to the $K_i$ level could reduce the conversion of D-fructose to D-fructose 6-phosphate by competitive inhibition in a dose-dependent manner. However, concentrations of D-tagatose above the $K_i$ would not have any additional inhibitory effect on fructokinase phosphorylation of D-fructose to D-fructose 6-phosphate. We hypothesize that these multiple chain-inhibitory effects on metabolism are probably significant enough to prevent formation of conidiophores and conidia; the fungicidal effects of D-tagatose treatment are similar to those caused by the synthesized fungicides (Fig. 6). These multiple chain-inhibitory effects on plural target sites in metabolic pathways should also be advantageous in reducing the risk of resistance developing in the pathogens. As we described above, these properties and its global approval as safe for foods are highly desirable for the use of D-tagatose as an agrochemical, and we have already developed a methodology and formulation to reduce the effective concentration of D-tagatose by about 6 times without affecting its efficacy[43,44].

Interestingly, D-tagatose was found in the root exudates of maize seedlings, and its level increased after treatment with humic acid, which is generated during the microbial decomposition of plant matter[45]. Twenty sugars including D-tagatose were exuded in larger amounts (>4-fold) at the root interface from plants treated with humic acid, and these exudates were predicted to influence microbial population size and community structure[45]. D-Tagatose was also identified as a component of a gum exudate of the cacao tree (*Sterculia setigera*)[46] and as a component of an oligosaccharide in lichens (*Rocella* species)[47]. If the release of D-tagatose as a plant exudate is induced by factors such as humic acid, then we speculate that the secreted D-tagatose might prevent glycolysis and mannose metabolism as a natural biobarrier against microbes around the plants. Although we did not find much involvement of plant defense systems in the D-tagatose-induced resistance to downy mildew, plants might have

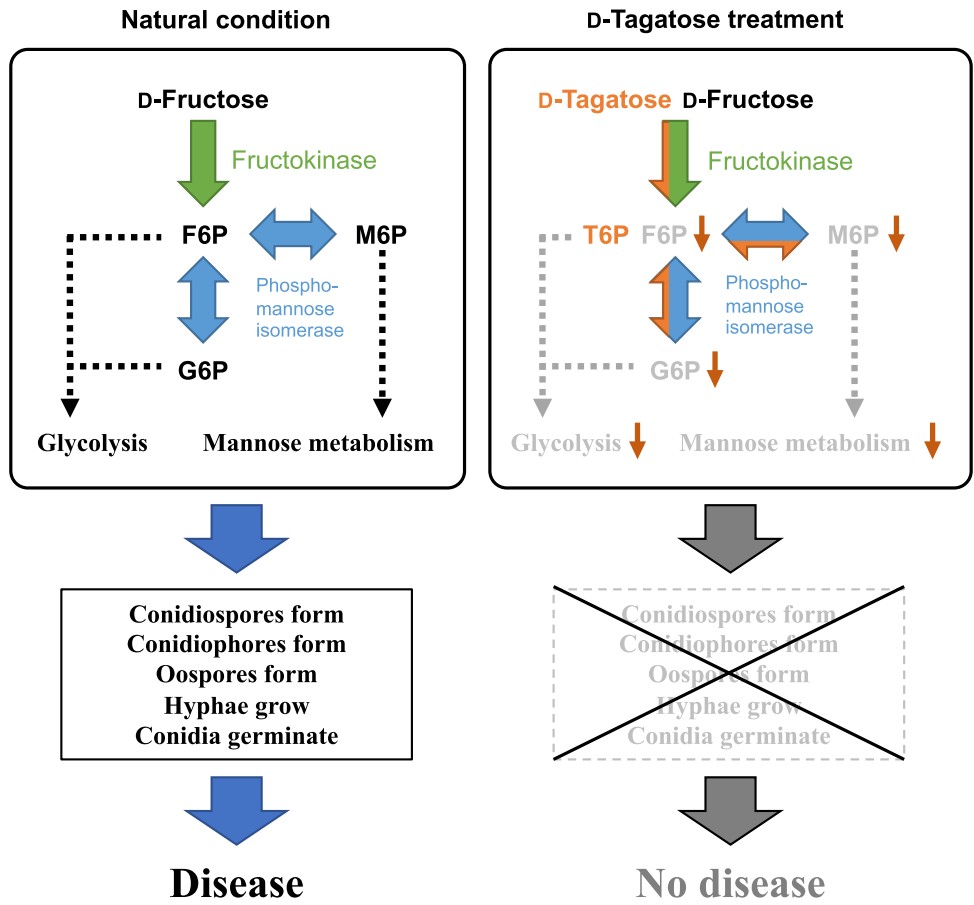

**Fig. 6 Hypothesis to explain D-tagatose effect on monosaccharide metabolism by fructokinase and phosphomannose isomerase in *Hyaloperonospora arabidopsidis*, causal agent of Arabidopsis downy mildew.** The inhibitory effects caused by D-tagatose are not a simple block of fructokinase (LC500344) activity, but rather a competitive inhibition at the maximum level ($K_i$), and the phosphorylated product, D-tagatose 6-phosphate (T6P), likely inhibits the metabolic enzyme phosphomannose isomerase (LC500563), which can produce both D-glucose 6-phosphate (G6P) and D-mannose 6-phosphate (M6P) from D-fructose 6-phosphate (F6P). This inhibition leads to a reduction of glycolysis and mannose metabolism in multiple developmental stages of *H. arabidopsidis*, including the formation of conidiophores, conidia, and oospores, and thus induces disease tolerance, and no symptoms develop.

alternative defenses against a pathogen weakened by D-tagatose. A lower concentration of D-tagatose was significant enough to prevent not only downy mildew but also powdery mildew. Unlike the case of downy mildew pathogens, typical components of the walls of powdery mildew pathogens are not known. Zeyen et al.[48] reported a suppressive effect by D-mannose on resistance to powdery mildew infection, but there may be different mode of actions of D-tagatose other than what we described for downy mildew and other pathogens. Thus, the mode of action of D-tagatose is neither entirely clear, nor is the role of D-tagatose in nature known. However, we believe our discovery of the effects of the rare sugar D-tagatose opens a gate for new types of agrochemicals to protect various agriproducts.

## Methods

**Chemicals**. Rare sugars and their derivatives (Supplementary Fig. 1) with respective purities of 100% were prepared by the Rare Sugar Production Station, International Institute of Rare Sugar Research and Education, Kagawa University, Japan using methods described previously[2,3,49]. Common sugars and other reagents described in the Methods section were purchased from Fujifilm-Wako (Tokyo, Japan) unless noted otherwise.

**Effects of D-tagatose on diseases in pot and field trials**. D-Tagatose was tested against diseases caused by various oomycete, ascomycete, or basidiomycete pathogens (Table 1) to determine the effective dose (percentage in weight per volume, w/v) that reduced disease severity to less than 50% of that caused by the mock treatment (untreated).

For pot trials, a control value was calculated based on disease severity after a spray of D-tagatose solution at 0.5–10% (w/v) in distilled water containing New Gramin sticker (3 ml 10 liter$^{-1}$) (Mitsui Chemicals Agro Inc., Tokyo, Japan) compared to severity without the D-tagatose spray by methods described previously[43,44,50] (Experimental conditions for the respective host–pathogen combinations for pot trials in Table 1 are summarized in Supplementary Table 1).

Typical disease symptoms and less-severe symptoms after other treatments for cucumber downy mildew are shown as examples in Fig. 1a. A solution (5 ml per pot) of 1 or 5% (w/v) D-tagatose, D-allulose, or D-allose was sprayed on the first true leaf of cucumber seedlings (cv. Sagami–Hanshiro; 1- to 1.2-leaf stage) 5 days before inoculation with 5 ml ($2 \times 10^4$ sporangia ml$^{-1}$) per pot of downy mildew pathogen (*Pseudoperonospora cubensis*, FS-1 strain [stock isolate of National Federation of Agricultural Cooperative Associations]). Inoculated plants were incubated at 20 °C in a moist chamber under 100% humidity for 24 h. A solution (5 ml) of 0.01% (w/v) probenazole [PBZ], 0.005% (w/v) acibenzolar-S-methyl [ASM], or 0.03% (w/v) metalaxyl was also sprayed on the first true leaf of cucumber seedlings (cv. Sagami–Hanshiro) 5 days before inoculation with downy mildew pathogen (5 ml, $2 \times 10^4$ sporangia ml$^{-1}$) as described above. These plants and untreated plants without any sugar or agrochemical treatment were incubated for 7 days in a greenhouse at 20 °C. The timing of the treatments with sugars and agrochemicals also varied from 1 to 7 days before inoculation and from 12 to 72 h after inoculation (5 ml, $1 \times 10^4$ sporangia ml$^{-1}$). Symptoms were then assessed after 7 days under the conditions described above. Severity was calculated from a rating 0–3: 0, no symptoms; 0.1, diseased area 3%; 0.3, 10% diseased area; 0.8, 25% diseased area; 1.5, 50% diseased area; 2, 70% diseased area; 3, 95% or more diseased area (see also Supplementary Table 1) as described previously[43,44,50]. The relative disease severity was calculated against that caused for the untreated plants based on the average (±SD) ($N = 3$ independent replications), then the data from the respective curative or preventive experiment were analyzed using a Tukey–Kramer multiple comparison test ($p < 0.05$) in the program JMP 12 (SAS Institute, Cary, NC, USA) (Fig. 1b).

In field trials, the control efficacy of 1 or 5% (w/v) D-tagatose against five downy mildew–host combinations (grapevine cv. Kyoho caused by *Plasmopara viticola*, cucumber cv. Ancor 8 by *Pseudoperonospora cubensis*, Chinese cabbage cv. Kigokoro75 by *Hyaloperonospora parasitica*, onion cv. Gifuki by *Peronospora destructor*, or spinach cv. Jiromaru by *Peronospora farinosa* f. sp. *spinaciae*) was reexamined and compared with that on plants treated or untreated with an appropriate reference fungicide: cyazofamid FL (CZF; Ranman, Ishihara Sangyo Kaisha, Osaka, Japan), chlorothalonil FL (CTN; Daconil, SDS Biotech K. K., Tokyo, Japan), or mancozeb WP (MCZ; Dithane, Corteva Agriscience, Wilmington, DE, USA) based on the average of disease severity (±SD) ($N = 50$–163 leaves per replication, three independent replications). Since effective concentrations for the respective fungicides differ against the specific downy mildews, CZF was used at 94 parts per million (ppm) against grapevine and spinach downy mildew, CTN at 400 ppm against Chinese cabbage and cucumber downy mildew, and MCZ at 1875 ppm against onion downy mildew. All isolates of the pathogens, except those from natural infections, were obtained from standard stocks of the Agrochemical Research Center, Mitsui Chemicals Agro, Inc. and maintained in the laboratory as appropriate.

Inoculation of the pathogens and treatment of D-tagatose or respective agrochemicals were summarized in Supplementary Table 2. Disease severity was calculated from the rating based on the Fungicide Evaluation Manual of Japan Plant Protection Association for field trials[51] (also see Supplementary Table 2) as described previously[44,50]. Briefly, plants grown until a specific Biologische Bundesanstalt, Bundessortenamt und CHemische Industrie (BBCH) stage[52], were sprayed with 200 to 300 liters D-tagatose solution per 10 a with a sprayer. As one of examples, sporangia of *Plasmopara viticola* ($1 \times 10^3$–$10^4$ sporangia ml$^{-1}$) were sprayed 2 days after the D-tagatose or agrochemical treatment, and the treatment of D-tagatose or agrochemical was repeated two more times with about one week interval. The disease severity of the tested plants was scored in accordance with the following criteria 7 days after the final treatment (Supplementary Table 2). The disease severities shown in Fig. 1c–g were calculated based on a disease rating specific for each disease and based on percentage diseased leaf area or number of lesions (e.g., grapevine downy mildew: 0, no symptoms; 1, area <10%; 2, area 11 to 30%; 3, area 31–50%; and 4, area ≥ 51%) (Supplementary Table 2). Respective disease ratings were used to calculate disease severity as $100[(n_1 + 2n_2 + 3n_3 + 4n_4)/4 N]$; $N$ = total number tested (50–163 leaves per replication, with three replications to calculate the average), $n_1$–$n_4$ = number of leaves rated, respectively as disease rating 1–4 for grapevine, cucumber, onion and spinach, and $100(n_1 + 2n_2 + 3n_3)/3 N$; $N$ = total numbers tested (160 leaves per replication, with three replications to calculate the average), $n_1$–$n_3$ = number of leaves for each disease rating 1–3 for Chinese cabbage. The tests were repeated three times for each treatment, and the average disease severity ±SD was calculated and is shown in Fig. 1. All data in Fig. 1 were analyzed using a Tukey–Kramer multiple comparisons ($p < 0.05$) in the program JMP 12. Experimental conditions for other host-pathogen combinations for field trials in Table 1 are also summarized in Supplementary Table 2.

For testing D-tagatose effects on downy mildew of Arabidopsis (Fig. 2a), seedlings of *A. thaliana* ecotype Colombia-0 (Col-0), grown in soil in plastic pots (7.5 cm diameter) at 22 °C with 16 h light (100–150 μmol m$^{-2}$ s$^{-1}$)/8 h dark for 3 weeks, were sprayed with or without 10 ml of 1% (w/v) aqueous D-tagatose per pot. After 1 day in the same conditions, the plants were sprayed with $1 \times 10^5$ conidiospores ml$^{-1}$ of *Hyaloperonospora arabidopsidis* isolate Noco2, and then incubated at 18 °C and 90–100% relative humidity with 10 h light (150–200 μmol m$^{-2}$ s$^{-1}$)/14 h dark. Plants were photographed at 0, 10, or 20 days post inoculation (dpi) (Fig. 2a, b).

For the tests of D-tagatose effects on downy mildew of Arabidopsis described in Fig. 2c–h, seedlings of *A. thaliana* ecotype Col-0 were grown on a filter paper (Whatman 3MM Chr) (GE Healthcare, Chicago, IL, USA) on Murashige–Skoog medium with 1% (w/v) sucrose and 0.8% (w/v) agar at 22 °C with 16 h light (50–200 μmol m$^{-2}$ s$^{-1}$)/8 h dark for 10 days then transferred to Murashige–Skoog medium (2 ml) containing D-tagatose (up to 100 mM) or 50 mM of deoxygenated D-tagatose at the C-6 position (6-deoxy-D-tagatose, 6dT) (Supplementary Fig. 1). Seedlings on the paper filter were kept for 1 day in the same conditions, and then the seedlings were sprayed with $1 \times 10^5$ conidia ml$^{-1}$ of *Hyaloperonospora arabidopsidis* isolate Noco2 and incubated at 18 °C and 90–100% relative humidity with 10 h of light (150–200 μmol m$^{-2}$ s$^{-1}$)/14 h dark for 7 days.

The extent of hyphal growth of *H. arabidopsidis* isolate Noco2 7–10 days after inoculation of Arabidopsis seedlings with or without D-tagatose treatment was examined microscopically after trypan blue (0.01% w/v in lactophenol) staining with chloral hydrate destaining[36]. Typical hyphal growth of *H. arabidopsidis* isolate Noco2 in Arabidopsis seedlings after treatment with or without 1% (w/v) D-tagatose treatment were imaged at 10 days after inoculation (Fig. 2b). Hyphal growth on 111–138 cotyledons for each treatment 7 days after inoculation was rated based on percentage of leaf area with hyphae: +++ for 75–100% of entire leaf; ++ for 25% to 74%; + for <24%; and – for no hyphae. Typical hyphal growth for each rating, classified as hyphal growth efficiency (%), is shown in Fig. 2c as means ± SD of three independent replications.

Conidiospores produced by *H. arabidopsidis* isolate Noco2 on 20 (Fig. 2d) or 30 (Fig. 2h) Arabidopsis seedlings with four leaves including cotyledons were collected in water by vortexing at 7 days after inoculation for each treatment described above with or without D-tagatose treatment (up to 100 mM) or 50 mM of 6dT, then

counted with a hemocytometer. Each treatment was done six times, and the mean number of conidia per 20 seedlings (±SD) per treatment (Fig. 2d) or per 10 seedlings (±SD) per treatment (Fig. 2h) of six independent replications was calculated.

Conidiophores and oospores produced by *H. arabidopsidis* isolate Noco2 on or in each cotyledon ($N = 37$–49) were observed microscopically and counted for each treatment at 7 days after the inoculation described above with D-tagatose treatment up to 100 mM or without. The tests were repeated three times for each treatment, and means (±SD) per cotyledon calculated (Fig. 2e, f).

Typical conidiospores, conidiophores and oospores observed without D-tagatose treatment are shown in Fig. 2g. All data in Fig. 2c–h were analyzed using a Tukey–Kramer multiple comparison test ($p < 0.05$) in the program JMP 12.

**In vitro effects of D-tagatose on *Hyaloperonospora arabidopsidis* isolate Noco2.** To test direct effects on germination and germ tube elongation 6 h after germination, we collected fresh conidiospores of *H. arabidopsidis* isolate Noco2 and placed 10 μl of $2 \times 10^4$ conidiospores ml$^{-1}$ in each well of a 96-well microtiter plate (Falcon no. 3075; NJ, USA) that held 100 μl Gamborg B5 medium salt mixture solution (Nihon Pharmaceutical, Tokyo, Japan) and 10 μg/ml meropenem with or without D-tagatose or D-mannitol up to 400 mM in each well. The plate was then covered with parafilm and incubated for 48 h (total 54 h) at 12 °C in the dark. Ten microliters of $2 \times 10^4$ conidia ml$^{-1}$ was also placed on Gamborg B5 medium salt mixture solution (Nihon Pharmaceutical) with or without 50 mM D-tagatose or 6-deoxy-D-tagatose in microtiter plates and incubated the same way.

Typical germ tubes in each test were photographed (Fig. 2i), and their lengths were measured 48 h after the sugar was added (at 6 h after germination; 54 h total incubation); mean length ± SD ($n = 60$–93 independent germ tubes) for each treatment is shown in Fig. 2j, and the mean percentage germination ± SD ($n = 85$–263 for each of 4–6 independent replications) at 24 h is shown in Fig. 2k. Means among treatments were compared for significant differences using a Tukey–Kramer multiple comparison test ($p < 0.05$) in JMP 12.

**RT-qPCR, microarray, and RNA-seq analyses of regulation of gene expression by D-tagatose.** For RT-qPCR analyses of PR-protein and defense-related gene expression in cucumber, seeds of cucumber (*Cucumis sativus* L. cv. Sagami–hanshiro) were germinated in the dark on a filter paper (Whatman 3MM Chr) moistened with water. After 2–3 days, seedlings were planted in plastic pots (7.5 cm diameter) and grown in a growth chamber at 22 °C for 2 weeks in natural light in July (Kagawa, Japan). Immediately after plants were sprayed with or without 10 ml of 1% (w/v) aqueous D-tagatose per pot (7.5 cm diameter) and air-dried, they were sprayed with or without $1 \times 10^5$ conidia ml$^{-1}$ of *Pseudoperonospora cubensis* per pot, and all leaves from the plant were sampled at 0, 24, and 48 h after inoculation. The leaves were ground to a fine powder in liquid nitrogen with a mortar and pestle. Total RNA was extracted using Tri Reagent (Sigma, St. Louis, MO, USA). The cDNAs for cucumber genes were prepared with a PrimeScript RT Master mix (Takara, Shiga, Japan), and expression of several PR-protein genes was quantified using RT-qPCR with SYBR Premix Ex Taq II (Takara) and a Thermal Cycler Dice TP800 (Takara). Cycling conditions were as follows: 95 °C for 30 s, followed by 40 cycles of 95 °C for 30 s and 60 °C for 30 s. The transcript level was normalized by comparison with actin (AB010922), and data were analyzed as described previously[14,17]. Means with SD values (2 to 3 independent replications) were analyzed for significant differences among treatments using a Tukey–Kramer multiple comparison test ($p < 0.05$) using JMP 12 (Fig. 3a). For selecting target defense-related genes examined above, the same cDNAs from cucumber leaves treated with or without D-tagatose for 24 h under the above conditions were amplified using a Clontech PCR Select cDNA Subtraction Kit (Takara), and 288 random clones that were expected to be upregulated in D-tagatose-treated leaves were sequenced (Supplementary Table 3). Clones with annotated sequences for putative defense-related genes among those 288 clones were selected, and a partial region of each sequence was used for this RT-qPCR analysis as the putative PR-protein and defense-related genes encoding peroxidase (POX)[24], lipoxygenase (LOX)[25], pore-forming toxin-like protein (Hrf)[26], 4-coumarate-CoA ligase (4CL)[27,28], and caffeoyl-CoA O-methyl-transferase (CCoAMT)[27,28]. Primers used for these analyses are listed in Supplementary Table 4.

For comparisons of rice PR-protein gene expression after various sugar treatments based on microarray analyses, an Agilent Rice Oligo Microarray (44k, custom-made; Agilent Technologies, Redwood City, CA, USA) was used with the methods described previously[16]. Briefly, seedlings of two-leaf-stage rice plants were cultured in Kimura B broth containing 0.5 mM of D-tagatose, D-allulose[14], D-allose[16,17], D-glucose[14,16,17], or no-sugar for 2 days. Total RNA was extracted using an RNeasy Plant Mini Kit (Qiagen, Hilden, Germany). The RNAs (400 ng) were labeled with Cy3 or Cy5 using a Low RNA Input Linear Amplification/ Labeling Kit (Agilent Technologies) after hybridization and washing according to the instructions. Hybridized microarrays were then scanned with an Agilent Microarray Scanner (Agilent Technologies), and signal intensity of each spot in the array was delineated, measured and normalized using Feature Extraction Software (version 9.1; Agilent Technologies). The microarray analyses were done three times. Scatter plot analyses, done using the Subio platform 1.22.5473 (Subio, Amami, Japan), are shown in Supplementary Fig. 3. Data extraction processes were performed using three independent replications for each treatment according to

the instructions, and relative expression of the respective PR-protein genes was calculated with SD values in relation to data from the no-sugar treatment. The relative expression values were compared using a Tukey–Kramer multiple comparison test ($p < 0.05$) using JMP 12 in Fig. 3b. All microarray data were deposited as data files in the Gene Expression Omnibus Database with accession GSE19595 for D-allulose[14], GSE15479 for D-allose[16,17] and D-glucose[14,16,17], and GSE136313 for D-tagatose.

For RNA-seq analyses of D-tagatose effects on downy mildew-inoculated Arabidopsis (Fig. 3c, Supplementary Fig. 4), seedlings of *A. thaliana* ecotype Col-0 grown in soil in plastic pots (7.5 cm diameter) at 22 °C with 16 h light (100–150 μmol m$^{-2}$ s$^{-1}$)/8 h dark for 10 days were sprayed either with or without 10 ml of 1% (w/v) aqueous D-tagatose per two plastic pots (7.5 cm diameter) and kept for a day in the same conditions. After the sugar treatment, Arabidopsis plants were sprayed with $1 \times 10^6$ conidia ml$^{-1}$ of *H. arabidopsidis* isolate Noco2, then incubated at 18 °C and 90–100% relative humidity with 10 h light (150–200 μmol m$^{-2}$ s$^{-1}$)/14 h dark for another 6 days. Twenty to thirty biological replicates of plants with conidiospores and conidiophores that were inoculated and treated with or without D-tagatose were ground to a fine powder with a mortar and pestle in liquid nitrogen. Total RNA was extracted using Tri Reagent (Sigma), then purified using an RNeasy Plant Mini Kit (Qiagen). The quality and quantity of the extracted RNA was determined using the Nanodrop Spectrophotometer (Thermofisher) and Agilent 2100 Bioanalyzer (Agilent Technologies). The A260/A280 values and the 28 S/18 S ribosomal RNA ratios of each total RNA sample were 2.06 and 1.7 (D-tagatose-treated sample), and 2.13 and 1.6 (mock-treated sample), respectively. The purified total RNA was sent to Hokkaido System Science (Sapporo, Japan) for RNA-seq. The library for RNA-seq analyses was prepared with 5 μg total RNA using the TruSeq RNA Sample Prep Kit v2 (Illumina, San Diego, CA, USA) and the manufacturer's instructions. The libraries were sequenced using the HiSeq 2000 sequencing platform and TruSeq SBS Kit v3 reagents with 101 cycles. Base-calling and data filtering were performed using CASAVA ver.1.8.1 software (Illumina), ultimately yielding approximately mapped reads or 122 million (D-tagatose-treated sample) or 84 million (mock sample) bases. The mean quality scores for the reads were 35.8 (D-tagatose-treated sample; 91.81% had ≥Q30 bases) and 35.57 (mock sample; 91.18% had ≥Q30 bases). The adapter sequences were trimmed by Cutadapt (v1.1)[53]. After preprocessing, the reads were mapped to the *A. thaliana* genome (TAIR10.17) and *H. arabidopsidis* Emoy2 genome (HyaAraEmoy2_2.0) from the Ensembl database (http://ensemblgenomes.org) using TOPHat (v2.0.2)/Bowtie[54]. Expression levels of each gene in both D-tagatose-treated and mocked samples were quantified by the number of fragments (paired-end reads) mapped to the coding region of each gene with a value of FPKM (fragments per kilobase of exon per million fragments mapped) using Cuffdiff (v2.0.2) program[54]. Differential expression between D-tagatose-treated and mocked samples was calculated as a Log2 ratio of respective FPKM values. Statistical significance (q-value) of the comparison using FPKM values was also calculated using Cuffdiff (v2.0.2) program[54]. All data, including raw sequence files for samples, were deposited in the Gene Expression Omnibus Database as accession GSE136568.

Selected genes for the differential expression analyses were PR-protein and defense-related genes jasmonic acid (JA) and/or ethylene (ET)-signaling genes (JA/ET [PR-proteins]: PR-3/CHIB [At3g12500], PR-4 [At3g04720], PDF1.2a [At5g44420], PDF1.2b [At2g26020]; JA markers: LOX2 [At3g45140], VSP2 [At5g24770], JAZ1 [At1g19180], JAZ5 [At1g17380], JAZ10 [At5g13220], TPS4 [At1g61120], CLH1 [At1g19670]; ET markers: ERF1 [At3g23240], ORA59 [At1g06160], EIN2 [At5g03280], EIN3 [At3g20770]), salicylic acid (SA)-signaling genes (SA-related PR-proteins: PR-1 [At2g14610], PR-2 [At3g57260], PR-5 [At1g75040]; SA markers: EDS1 [At3g48090], ICS1 [At1g74710], TGA2 [At5g06950], TGA5/OBF5 [At5g06960], TGA6 [At3g12250]); MAMPs response genes (chitin/peptidoglycan/flagellin receptor genes: CERK1 [At3g21630], CEBiP/LYM2 [At2g17120], LYM1 [At1g21880], LYM3 [At1g77630], FLS2 [At5g46330], BAK1 [At4g32430]), and cell wall modification-related genes: cellulose synthase/polygalacturonase inhibitor (PGIP)/pectin methyl esterase (IRX1/CESA8/LEW2 [At4g18780], IRX3/CESA7 [At5g17420], IRX5/CESA4 [At5g44030], PGIP1 [At5g06860], PGIP2 [At5g06870], ATPMEPCRA [At1g11580]), respectively. Differential expression of these genes between D-tagatose-treated and mocked samples was summarized in Supplementary Fig. 4a, and the values of log2 ratio shown in Supplementary Fig. 4a were visualized using graduated color bars (Fig. 3c). All genes up- or downregulated more than 10 times from either *A. thaliana* or *H. arabidopsidis* isolate Noco2 were also listed with the annotation in Supplementary Fig. 4b.

### Detection of monosaccharide kinase activity in crude enzyme extracts from germinated conidia of *Hyaloperonospora arabidopsidis* isolate Noco2. Monosaccharide kinase activity of crude enzyme extracts from germinated conidia of *H. arabidopsidis* isolate Noco2 was determined. Germinated conidiospores ($2 \times 10^6$ conidiospores) that had been grown for 24 h in Gamborg B5 medium salt mixture solution (Nihon Pharmaceutical) were ground with a mortar, pestle and zirconia beads (0.5 mm diameter) in 500 μl of extraction buffer (50 mM Tris-HCl buffer, pH 7.5, containing EDTA-free Complete protease inhibitor [Sigma]), then centrifuged at 13,000 rpm for 5 min at 4 °C. The supernatant was desalted using 50 mM Tris-HCl buffer (pH 7.5) and an Amicon Ultra-4 (10,000 MWCD;

Millipore, Billerica, MA, USA), and the supernatant (=crude enzyme extracts) was mixed with 100 mM of D-tagatose in the reaction mixture (50 mM Tris-HCl pH 7.5, 10 mM MgCl$_2$ and 6 mM ATP). The mixture was incubated for 24 h at 25 °C, and the reaction products were labelled using the *p*-aminobenzoic acid ethyl ester (ABEE) system (J-Chemical, Tokyo, Japan) and detected using HPLC as detail in the section on characterization of enzymes for D-tagatose phosphorylation using HPLC.

### cDNA cloning and heterologous expression of various genes encoding sugar kinases. A cDNA library of *H. arabidopsidis* isolate Noco2 was constructed using total RNA (1 μg) extracted as described for the RNA-seq analyses and a SMARTer PCR cDNA Synthesis Kit (Takara) according to the instructions. cDNA fragments of the coding region from fructokinase (LC500344), glucokinase (LC500564), xylulose kinase (LC500562), or ribokinase (LC500561) of *H. arabidopsidis* isolate Noco2 were subcloned in frame into the pGEX5X-1 vector (GE Healthcare) and expressed in SoluBL21 Competent *E. coli* cells (Genlantis, San Diego, CA, USA) according to the manufacturer's instructions. Primers used for cloning are listed in Supplementary Table 4. The recombinant proteins of each coding region, generated by the heterologous expression system with 0.4–0.6 mM Isopropyl β-D-1-thiogalactopyranoside (IPTG) for 3 h at 37 °C, were purified using a GSTrap HP column (GE Healthcare) and the manufacturer's instructions. Images of the recombinant proteins separated using SDS-PAGE with a standard protocol[16] are shown in Fig. 4b and Supplementary Fig. 5, and the prepared recombinant proteins were used for initial examination of their D-tagatose phosphorylation activities (Fig. 4c, Supplementary Fig. 5).

For enzymatic characterizations, the recombinant protein of fructokinase (LC500344) was generated using a mass culture system with 10 liter LB broth containing 100 μg ml$^{-1}$ carbenicillin at 37 °C and 200 rpm for 6 h (preculture), and 25 °C and 200 rpm for 1 day with 0.6 mM IPTG in a Jar Fermenter TS-M15L (Takasugi MFG, Tokyo, Japan) and purified using a GSTrap HP column (GE Healthcare) as per the instructions and dialyzed against 5 mM Tris-HCl buffer (pH 7.5) for 4 h. The prepared recombinant protein was used to determine phosphorylation activity for a kinetics analysis (Fig. 4d, Supplementary Fig. 6a–d, j, k), optimal temperature (Supplementary Fig. 6e), thermal stability (Supplementary Fig. 6f), optimal pH (Supplementary Fig. 6g), pH stability (Supplementary Fig. 6h), and optimal cofactor (metal ion) (Supplementary Fig 6i).

### Characterization of enzymes for D-tagatose phosphorylation using HPLC. The enzymatic reaction mixture (1 ml containing 100 mM Tris-HCl (pH 7.5), 10 mM MgCl$_2$, 6 mM ATP, 25 mM D-tagatose or D-fructose, and 100 ng target enzymes) was incubated for 24 h at 25 °C. For detecting kinase products using HPLC, ABEE labelling was performed using the method of Yasuno et al.[55] with modifications[16]. Briefly, 10 μl of the reaction mixture or sugar standard (25 mM D-fructose, D-tagatose, D-fructose 6-phosphate, or D-tagatose 6-phosphate) was added to 40 μl of ABEE reagent solution with a borane–pyridine complex (from the kit) and heated at 80 °C. Chloroform and distilled water (200 μl each) were added, the mixture centrifuged at 3000 rpm for 5 min. The upper aqueous layer was used for HPLC analyses (Prominence; Shimadzu, Kyoto, Japan) using an Xbridge C18 column (4.6 mm ID × 250 mm; Waters, Milford, MA, USA) with a 50-min separation at a flow rate of 1.0 ml min$^{-1}$ at 30 °C with a running solvent system of 0.2 mM of potassium borate buffer (pH 8.9)/acetonitrile (93/7), followed by a 20-min wash with 0.02% trifluoroacetic acid/acetonitrile (50/50) and equilibration for 15 min with the running solvent. The peaks were monitored with a fluorescence detector (RF-10A XL or RF-20A XL, Shimadzu) with emission of 360 nm and excitation of 305 nm or UV detector (UV-VIS Detector SPD-10AV, Shimadzu) at an absorbance of 305 nm.

### Inhibition of monosaccharide phosphorylation activity of fructokinase by D-tagatose. Activity of fructokinase (10 μg) was determined spectrophotometrically at 340 nm at 25 °C for 5 min by coupling production of ADP to oxidation of NADH via pyruvate kinase (15 U ml$^{-1}$) (Oriental Yeast, Tokyo, Japan) and lactate dehydrogenase (10 U ml$^{-1}$) (Oriental Yeast) reactions as described by Miller and Raines[56] with 0.5–2.5 mM D-fructose and 10–250 mM D-tagatose in 1 ml reaction mixture containing 80 mM Tris-HCl (pH 7.5), NADH (0.3 mM), phosphoenolpyruvate (0.8 mM), ATP (4 mM), and MgCl$_2$ (8 mM). The same data (mean ± SD of four replications of the reaction) were used for calculations for the kinetic analyses including Dixon plot (Fig. 4d), Lineweaver–Burk plot (Supplementary Fig. 6j), and bar graph (Supplementary Fig. 6k).

### Characterizations of enzymes in D-fructose 6-phosphate metabolic pathway and inhibition of activity by D-tagatose 6-phosphate. The cDNA fragment (described above) of the coding region from phosphomannose isomerase (LC500563) of *H. arabidopsidis* isolate Noco2 was subcloned in frame into the pColdI vector (Takara), and expressed in *E. coli* SHuffle Express Competent cells (New England BioLabs, MA, USA) according to the manufacturer's instructions. Primers used for cloning are listed in Supplementary Table 4. The recombinant proteins of the coding region generated by the heterologous expression system were obtained using a mass culture system with 10 liters LB broth containing 1% (w/v) D-glucose and 100 μg ml$^{-1}$ carbenicillin at 30 °C and 200 rpm for 6 h (preculture),

15 °C and 200 rpm for 1 h (cold shock induction), and 15 °C and 200 rpm for 3 days in Jar Fermenter TS-M15L and purified using a HisTrap HP column (GE Healthcare) as per the manufacturer's instructions, then dialyzed against 10 mM Tris-HCl buffer (pH 7.4). The recombinant proteins, separated by SDS-PAGE using a standard protocol[16], are shown in Fig. 5b.

A reaction mixture (100 µl) containing 100 mM Tris-HCl (pH 7.4), 10 mM MgCl₂, 10 µg recombinant protein of phosphomannose isomerase (LC500563), and 2.5 mM D-fructose 6-phosphate (F6P) was incubated at 30 °C for 6 h (Fig. 5c), or reaction mixtures (100 µl) containing 100 mM Tris-HCl (pH 7.4), 10 mM MgCl₂, 5 µg recombinant protein with 2.5 mM of F6P and D-tagatose 6-phosphate (T6P) (from 0 to 37.5 mM) were incubated at 30 °C for 1 h. Reaction products were labelled and purified for ABEE as described above, and the ABEE-labelled sugars (10 µl) were analyzed using an HPLC system (Prominence) with an Xbridge C18 column (4.6 mm ID × 250 mm) and 20-min separation at a flow rate of 1.0 ml min⁻¹ at 30 °C with solvent A of the Glyscope solvent set (J-Chemical), followed by a 5-min wash with solvent B from the solvent set and equilibration for 10 min with the running solvent. The peaks were monitored with the fluorescence detector (RF-20A XL) with emission of 360 nm and excitation of 305 nm. Respective data in Fig. 5d are means ± SD of three replications calculated from the peak area on the HPLC spectrum with a calibration curve using standards with known concentrations, and they were compared for significant differences using a Tukey–Kramer multiple comparison test ($p < 0.05$) in JMP 12.

**Statistics and reproducibility**. All data in this work were collected in multiple and respective data are means ± SD of replications and statistical analyses shown in figures were performed using a Tukey–Kramer multiple comparison test ($p < 0.05$) in JMP 12 program (SAS Institute) unless noted otherwise as described above.

**Reporting summary**. Further information on research design is available in the Nature Research Reporting Summary linked to this article.

## Data availability
All nucleotide sequences were deposited in the DNA Data Bank of Japan (DDBJ) as accession numbers LC500344 (fructokinase), LC500561 (ribokinase), LC500562 (xylulose kinase), LC500563 (phosphomannose isomerase), and LC500564 (glucokinase). Microarray and RNA-seq datasets in this work have been deposited in the NCBI Gene Expression Omnibus (GEO) under accession number GSE136313 (microarray) and GSE136568 (RNA-seq). Source data are available as Supplementary Data 1. All other data that support the findings of this study are available from the corresponding authors upon reasonable request.

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

## Acknowledgements

This study was supported in part by projects of the Bio-oriented Technology Research Advancement Institution of the Programme for Promotion of Basic and Applied Researches for Innovations in Bio-oriented Industry and the Research Program on Development of Innovative Technology of the National Agriculture and Food Research Organization (NARO), and the Regional Innovation Ecosystems of the Ministry of Education, Culture, Sports, Science and Technology, Japan. We are grateful to Dr. Hideki Takahashi (Tohoku University) for providing *Hyaloperonospora arabidopsidis* isolate Noco2; Dr. Yoshiaki Nagamura and Ritsuko Motoyama (NARO) and Dr. Yumiko Kokudo-Yamasaki (Kagawa University) for their support on the rice 44 K microarray analysis; Dr. Yoko Nishizawa (NARO), Dr. Takeo Okochi, Fumito Kasai, Dr. Hiromi Sano and Dr. Shigehiro Kato (Mitsui Chemicals Agro Inc.) for their valuable comments and suggestions; and Dr. Yutaka Ishida and Dr. Kazumasa Kakibuchi (Shikoku Research Institute Inc.) for information and discussion. Stock isolate FS-1 of *Pseudoperonospora cubensis* was kindly provided by the National Federation of Agricultural Cooperative Associations.

## Author contributions

S.M., T.F., T.O., K.O., Y.S., K.T., S.T., K.I., K.E., and K.A. designed and conceived the experiments; S.M., T.F., T.O., and K.O. performed the experiments, with the first three authors contributing equally; A.Y., K.I., K.G., and K.I. provided rare sugars and critical advice; S.M., T.F., T.O., and K.A. wrote, and all authors edited the final draft and revised the manuscript.

## Competing interests

T.F., T.O., Y.S., K.T., and K.E. are employed by Mitsui Chemical Agro, Inc. Remaining authors declare no competing interests.
