## [Peer Review File · Communications Biology]

Reviewers' comments:

Reviewer #1 (Remarks to the Author):

There is no doubt that agrochemicals play a key role in ensuring food production, and natural products are an important arsenal for discovering new pesticide molecules. In this manuscript, the rare sugar D-tagatose showed some activity against downy mildews, and the mode of action was also studied. The result showed that D-tagatose is not a plant defense activator, D-tagatose may be as a competitive inhibitor of D-fructose kinase activity, and D-Tagatose 6-phosphate sequentially inhibits phosphomannose isomerase. These effects block initial infection and structural development needed for reproduction. This work is reasonable and well-performed. Although D-tagatose showed poor fungicidal activity comparable to synthetic fungicides, these discoveries should open a new gate for developing safe, innovative fungicidal agrochemicals using a natural product for crop disease control essential for food production. Therefore this manuscript should be accepted for publication on Communications Biology.

Reviewer #2 (Remarks to the Author):

In the manuscript by Mochizuki et al., entitled "A sweet solution for an innovative, eco-friendly agrochemical using a rare sugar", the authors showed that the rare sugar D-tagatose, a safe natural product, controls a wide range of plant diseases for protection of agriproducts for food productions, especially on downy mildews disease control. They found the D-tagatose acts on the pathogen but not plant. They also proved that D-tagatose likely works as a competitive inhibitor of D-fructose kinase activity to inhibits the first step of mannose metabolism. The chain-inhibitory effects on metabolic steps are significant enough to block initial infection and structural development needed for reproduction such as conidiophore and conidiospore formation of downy mildew. In my opinion, this discovery indeed opens a new gate for developing safe, innovative fungicidal agrochemicals using a natural product for crop disease control for food production.

I raised some questions need to be addressed.

1. I concern about the effect of D-tagatose could be not high enough, for the effect concentration is relatively too high for wide field apply. The authors should discuss about this, and compared the effect with other types of disease controlling reagents. A 0.5% concentration could still be a high cost for utilization. Try to discuss the possibilities how to improve utilization.
2. The disease severity experiment design is not fine enough. The results of this part is relatively weak. It is necessary to test the disease control effect by treating the plant leaves with D-tagatose during different time points post pathogen inoculation. This is important, which could be an important clue to suggest whether D-tagatose can be used as prevention or recovery.
3. Fig.2a, no 0 h CK.
4. It is required to show classic disease symptoms on single leaf upon D-tagatose treatment.
5. It is required to show morphologies of conidiospores, conidiophores, and oospores upon D-tagatose treatment, compared with the normal condition.
6. For the discussion section is too long, it is very repetitive, described too much same to the results. This section should be more focus on the importance, but not repeat the result conclusion. The authors also need to discuss about situation of similar research or application field related.
7. Fig.S5, label problems. Fig.S6k, no error bar?

Reviewer #3 (Remarks to the Author):

The manuscript entitled 'A sweet solution for an innovative, eco-friendly agrochemical using a rare sugar' presented an interesting characterization of the mode of action of a rare sugar (tagatose) against a plant disease. The goal is interesting and experimental design is ambitious, but some aspects should be better considered in order to properly evaluate the impact and reproducibility of the obtained results.

The efficacy of tagatose against plant pathogens is largely known from several years (Ohara et al. 2008 Patent JP2008209921, PCT/JP2009/003925 and Ohara et al. 2014 Patent JP2013050518. PCT/JP2014/056181). Therefore, the manuscript should be better reconsidered according to the already known information available in the literature, in order to better highlight novel results of this work. In particular,

- known properties of tagatose against plant and human pathogens should be better reported in the introduction;
- the aim of the work (lines 76-78) should better focused on novel results, i.e. the mechanism of action of tagatose;
- already known information and citations should be removed from the Results session;
- Abstract and Discussion should be dramatically tuned down and better focalized on the real results reported in this work, that are based on the mechanism of action of tagatose, but not on efficacy against plant pathogens, since efficacy is already well known and patented by the authors of this manuscript,
- consequently, the Conclusion session (lines 337-356) appear too speculative and it is not properly linked to the presented results and the aims declared at the beginning of this manuscript.

Key controls seem not properly reported in the enzymatic assays. In particular:

- profiles of pure tagatose and tagatose-6-phosphates should be reported in supplementary Figures, in order to support the key HPLC results of this manuscript;
- since the recombinant fructokinase is a 'candidate' enzyme ('candidate' was properly declared by the authors) the assay of its activity in the phosphorylation of fructose should be reported as control. The activity of the candidate phosphomannose isomerase in producing mannose-6-phosphate was demonstrated in Figure 3b using fructose-6-phosphate as substrate. Therefore, a similar control should be performed on the candidate fructokinase, to demonstrate that the recombinant enzyme can produce fructose-6-phosphate using fructose as substrate in vitro. Formation of fructose-6-phosphated is mentioned at lines 298-301, without visible clear indication in the Figure 4.

The reproducibility of the data should be better clarified, in order to demonstrate robustness of the obtained results. I believe that authors appropriately carried out independent experimental repetitions with appropriate number of replicates. However, the number and type (single plant or pool of plants, single leaves or pool of leaves, tubes in case of in vitro assays) of biological replicates analyzed in each experiment should be better reported in the M&M session and in the legend of each Figure, in order to clarify how many values have been analyzed to build up the mean and standard deviation data reported in each Figure.

Likewise, each experiment should be repeated at least twice in two independent repetitions and reproducibility should be declared. This information should be provided for each experiment in the M&M and in the figure legends. In particular, how experimental repetitions are used in the data analysis should be reported in the manuscript, such as statistical tests to demonstrate that experimental repetitions are comparable should be declared. Moreover, it should be better declared if one representative experiment among N. repetitions is reported in each figure or if a pool of N. different experiments is averaged and reported in each figure.

This information is crucial to clarify reproducibility and possible variability of the results presented in

this work. Consequently, standard deviation values should be reported in each figure, Fig 2B, 3C, 4d, S4, S6i, S6j and S6k included. Description of number of biological replicates and independent experimental repetitions is particularly crucial for the enzymatic assays.

The Materials and Methods section is unclear and descriptions should be improved in order to easily allow the repetition of the experiments by other groups. Some key methods are reported in the supplementary materials (enzymatic assays), while other secondary methods (efficacy of tagatose against pathogens) are reported in a confused manner in the main document. If tagatose efficacy against pathogens is included in the manuscript, methods should be described appropriately. Descriptions are scarce and insufficient to repeat the experiment. Moreover, some key protocols cited in the manuscript are not accessible, such as conference proceedings of citation 50, or not easily understandable for an international scientific community, such as the manual written in Japanese of citation 51. Please consider that information to easily repeat each experiment for each assay against each pathogen should be provided for international scientific community. Therefore, descriptions should be improved and accessible citations should be mentioned. Protocols used to stain *H. arabidopsidis* in *Arabidopsis* leaves and to assess the disease severity are not appropriately described in the M&M. Moreover, the +, ++ and +++ classes should be converted to a more appropriate, and possibly objective, symptom class description. Likewise, description of direct effect assay on *H. arabidopsidis* (lines 462-471) is not clear and not supported by appropriate citations. Gene expression description is a mixture of different protocols, with missing parts on statistical analysis. On the other hand, some other parts include an excess in detail even though they are really standard protocols, such as the library preparation of Illumina which is well known. Please consider to avoid description of really standard protocols but improve descriptions of specific protocols used in this work, particularly on statistical analyses. For example, the statistical analysis to select the 288 clones upregulated in tagatose-treated leaves is missing and it should be clearly reported. A random selection is indicated, but cut off used to identify significant up-regulations should be provided. Likewise, statistical analysis used to compare mock-treatment and sugar-treatment on microarray results is missing. Once again, statistical analysis on RNA-Seq data is missing. DeSeq or similar statistical softwares are commonly used to detect significant up- and down-regulated genes in RNA-Seq analysis. Therefore, the number of replicates and statistical tests of RNA-Seq data should be reported in this manuscript.

Unclear descriptions can be found in the phytopathological part and this dramatically preclude to understand the confidence of the presented data. *H. arabidopsidis* is a well-known obligate biotroph, which grows only in presence of the alive host. Therefore, the *in vitro* protocol to germinate conidiophores in Gamborg B5 medium (lines 462-471 and 588-591) should be supported by appropriate references. In other words, no germination of an obligate biotroph is expected *in vitro*; the protocol of *H. arabidopsidis* *in vitro* seems not properly performed/described. Therefore, since *H. arabidopsidis* cannot grow *in vitro*, the cell amount used for the crude extract assay is probably really low. An estimation of the real quantity of tagatose-6-phosphate in the crude extract, expressed as micro gram per mg of fresh weight (or per mg of total proteins) is expected in order to clarify the quantification of Figure 4a.

Minor comments:

- Introduction describes several sugars and rare sugars, while it should be better focalized on tagatose properties and known information.
- Figures and Tables as usually cited in the Results, The used of Figure and Table citations in Introduction and discussion is discourage, to avoid confusions on novel data and data already reported in the bibliography.
- On the other hand, citations and introduction like sentences (e.g. lines 121-128, 136-144, 170-179,) should be avoided in the Results session.
- Some results are described in the M&M and figures are cited as well, and this dramatically reduces

the readability of this session.

- The M&M on enzymes characterization (lines 586-636) is a mixture of cDNA cloning and assays, I suggest to split it in different paragraphs to distinguish cloning session, assays on crude extracts and assays on purified recombinant proteins.
- Table 1 contains a mixture of published results and novel results, I suggest to mention published results in the introduction and Discussion, and to clarify better the novel results provided in this work.
- Four conditions are mentioned for Figure 3C (control, tagatose treatment and inoculation with the pathogen on both treatments), but only a single mean value is reported for each gene. All fold change values (with standard deviation) should be reported for each pairwise comparison, and significant changes should be highlighted.
- Statistical analysis is missing in Figure 3b, 3c and 4d.
- Results description of lines 12-147 should be improved, in order to better discriminate already published results from novel data. The use of citations and published results is preferred for the Discussion instead of Results session.
- Conclusion at lines 272-275 is not supported neither by appropriated references nor results of this manuscript.

Dear reviewers of Communications Biology,

Thank you very much for your favorable review and helpful criticism of our manuscript. The manuscript is now much better after the revision. Our responses to the advice and questions follows. The parts revised are in red font in the manuscript text, and our point of view on these respective revisions are described in parts I and II in this letter.

I: List organized by the Editor based on question/revise requests from reviewers, and our responses for respective items in the list

1. Treat the plant leaves with D-tagatose at different time points after pathogen inoculation to make the statement that it could be used as a preventative measure (Reviewer 2).

Thank you for this important point on whether D-tagatose works as a preventive or curative agent against disease development. Thus, we added data of Fig. 1 panel b, showing serial comparisons of disease severities on leaves treated with different concentrations (1 or 5%) of D-tagatose or chemical fungicide (metalaxyl) before or after inoculation of pathogen; relative values against those with no treatment are indicated. The results indicated that the tendency for inhibition of disease development was similar between D-tagatose and fungicide; no symptoms were observed in either case when plants were treated 5 days before inoculation, weak symptoms developed when plants were treated 7 days before the inoculation. On the other hand, symptoms started to be observed even when plants were treated with D-tagatose or fungicide 60 h after inoculation. Based on these results, we concluded that the D-tagatose effect is preventative and will provide very efficient control if plants are treated D-tagatose at least 5 days before pathogen attack, similar to the protection conferred by the fungicide (metalaxyl) treatment. Our data suggest that D-tagatose showed a 7-day residual effect similar to that of a fungicide effect, indicating that an application at 7-day intervals is likely to be of a practical use.

2. Provide classic disease symptoms on single leaf after D-tagatose as well as that of the different spore morphologies (Reviewer 2).

Thank you for another important point. We have added photo data on symptoms including Figs. 1a, 2a, 2b, and 2i. Typical symptoms of cucumber downy mildew are shown on an untreated plant, and after inhibition by fungicides (metalaxyl, ASM, and PBZ) and D-tagatose (1 and 5%) in Fig. 1a. The effects of D-allulose (=D-psicose) and D-allose described previously (e.g., Refs 14, 16, 17) are also shown as the control for rare sugars other than D-tagatose in Fig. 1a. However, no noticeable phytotoxicity was observed on plants treated with D-tagatose, while a significant inhibition on development of true leaf was observed on those treated with D-allose or D-allulose as a side effect with disease suppression due to stimulation of plant defense systems (e.g., Refs 14, 16, 17). As described in the text, unlike D-allulose and D-allose, D-tagatose is not a plant defense activator; it has direct inhibitory chain effects on mannose-related metabolic steps in pathogens, which is significant enough to block initial infection (Figs. 1 and 2). Images showing typical reproductive

structures are shown in Fig. 2g, and respective numerical data are given with results of statistical analyses (Fig. 2d-h) because no morphological changes were found for these structures in downy mildew.

3. Report the profiles of pure tagatose and tagatose-6-phosphates to support the HPLC (Reviewer 3).

Standard peaks of D-fructose, D-tagatose, D-fructose 6-phosphate, and D-tagatose 6-phosphate have been added to HPLC profiles in Fig. 4c. These molecules were labelled with p-aminobenzoic acid ethyl ester (ABEE: fluorescent labelling system to enhance the detection sensitivity). Two peaks (diastereomers) for each sugar were detected in respective HPLC charts because two labelled products were produced as described in the following structural figure.

Fig. Two peaks for D-fructose, D-fructose 6-phosphate, D-tagatose or D-tagatose 6-phosphate were detected in respective HPLC charts because of production of these two ABEE-labelled structures (diastereomers) for each sugar

4. Report the control for the fructokinase to demonstrate that it can produce fructose-6-phosphate (Reviewer 3).

HPLC profile of D-fructose 6-phosphate production by HaFK1 with D-fructose as the substrate was also added to Fig. 4c.

5. Make major reorganization and reframing of the manuscript as suggested by reviewer 3 and 2.

We revised the manuscript as suggested by reviewers as described in part II.

6. Provide details for some aspects in the methods (Reviewer 3).

We revised as suggested by reviewer 3 as described in part II.

7. All work should be associated with appropriate statistical analysis, with SD(s) provided, as well as the number of replicates and how the data was processed (Reviewer 3 provided detailed description of those points).

As detailed in part II, we revised with statistical analyses as suggested by reviewer 3.

8. Address all figure/textual/method clarifications and corrections pointed out by the reviewers.

As detailed in part II, we revised as suggested by the reviewers.

II: List of question/revise requests from reviewers and our responses

Reviewer #1 (Remarks to the Author):

There is no doubt that agrochemicals play a key role in ensuring food production, and natural products are an important arsenal for discovering new pesticide molecules. In this manuscript, the rare sugar D-tagatose showed some activity against downy mildews, and the mode of action was also studied. The result showed that D-tagatose is not a plant defense activator, D-tagatose may be as a competitive inhibitor of D-fructose kinase activity, and D-Tagatose 6-phosphate sequentially inhibits phosphomannose isomerase. These effects block initial infection and structural development needed for reproduction. This work is reasonable and well-performed. Although D-tagatose showed poor fungicidal activity comparable to synthetic fungicides, these discoveries should open a new gate for developing safe, innovative fungicidal agrochemicals using a natural product for crop disease control essential for food production. Therefore this manuscript should be accepted for publication on Communications Biology.

Thank you very much for your fair assessment and encouragement of our research. We aim to develop an innovative fungicidal agrochemical based on our discovery, and this paper will be the initial milestone for this process.

Reviewer #2 (Remarks to the Author):

In the manuscript by Mochizuki et al., entitled “A sweet solution for an innovative, eco-friendly agrochemical using a rare sugar”, the authors showed that the rare sugar D-tagatose, a safe natural product, controls a wide range of plant diseases for protection of agriproducts for food productions, especially on downy mildews disease control. They found the D-tagatose acts on the pathogen but not plant. They also proved that D-tagatose likely works as a competitive inhibitor of D-fructose kinase activity to inhibits the first step of mannose metabolism. The chain-inhibitory effects on metabolic steps are significant enough to block initial infection and structural development needed for reproduction such as conidiophore and conidiospore formation of downy mildew. In my opinion, this discovery indeed opens a new gate for developing safe, innovative fungicidal agrochemicals using a natural

product for crop disease control for food production.

Thank you very much for your fair assessment and encouragement of our research.

I raised some questions need to be addressed.

1. I concern about the effect of D-tagatose could be not high enough, for the effect concentration is relatively too high for wide field apply. The authors should discuss about this, and compared the effect with other types of disease controlling reagents. A 0.5% concentration could still be a high cost for utilization. Try to discuss the possibilities how to improve utilization.

There is always a balance between environmental risk and efficacy of a control agent. Although we described that symptoms caused by several downy mildews are inhibited by as low as 0.1% and more than 1% of D-tagatose inhibits to a level similar to that provided by fungicide treatments (Discussion section P8) and we have also developed methodology and formulation to reduce the effective concentration of D-tagatose by about 6 times without affecting its efficacy (Discussion section P11), some consumers might still consider the concentration to be a problem. However, considering recent regulation of synthetic chemical pesticides, we think these novel properties of D-tagatose and its global approval as safe for foods are highly desirable for use as an agrochemical. The multiple inhibitory chain effects on plural target sites in metabolic pathways might also be advantageous in reducing the risk of resistance developing in the pathogens as described in discussion section (P11).

2. The disease severity experiment design is not fine enough. The results of this part is relatively weak. It is necessary to test the disease control effect byvtreating the plant leaves with D-tagatose during different time points post pathogen inoculation. This is important, which could be an important clue to suggest whether D-tagatose can be used as prevention or recovery.

Thank you for the comment on the importance of determining whether D-tagatose works as a preventive or curative agent. We have added data (Fig. 1 panel b) showing serial comparisons of disease severity on leaves treated with 1 or 5% D-tagatose or chemical fungicide (metalaxyl) before or after inoculation of pathogen, with their relative values against those with no treatment. The results indicated that the tendency to inhibit disease was similar between D-tagatose and fungicide; no symptoms were observed when either one was applied 5 days before inoculation, while weak symptoms developed when these agents were applied 7 days before inoculation. On the other hand, symptoms began to develop even when D-tagatose or fungicide was applied 60 h after inoculation. Based on these results, we concluded that D-tagatose effect is preventative and will be efficient enough when applied at least 5 days before pathogen attack to the host plant, similar to the fungicide (metalaxyl) treatment. Our data suggest that D-tagatose has a 7-day residual effect similar to the effect of the fungicide, indicating that applications at 7-day interval application may be possible for a practical use, as we described also in part 1.

3. Fig.2a, no 0 h CK.

Thank you for the comment. We added a 0 h photos for both D-tagatose treatment and its control (Mock).

4. It is required to show classic disease symptoms on single leaf upon D-tagatose treatment.

Thank you for the comment. Although we answered this in part 1, we will copy/paste the response here. We have added photo data on symptoms including Figs. 1a, 2a, 2b, and 2i. Typical symptoms of cucumber downy mildew are shown on an untreated plant, and after inhibition by fungicides (metalaxyl, ASM, and PBZ) and D-tagatose (1 and 5%) in Fig. 1a. The effects of D-allulose (=D-psicose) and D-allose described previously (e.g., Refs 14, 16, 17) are also shown as the control of rare sugars other than D-tagatose in Fig. 1a. However, no noticeable phytotoxicity was observed on plants treated with D-tagatose, while a significant inhibition on development of true leaf was observed on that treated with D-allose or D-allulose as the side effect together with disease suppression due to stimulation of plant defense systems (e.g., Refs 14, 16, 17). As described in text, not like D-allulose and D-allose, D-tagatose is not plant defense activator and gives chain-inhibitory effects on mannose-related metabolic steps of pathogens directly, which is significant enough to block initial infection (Figs. 1 and 2). Images showing typical reproductive structures are shown in Fig. 2g, and respective numerical data are given with results of statistical analyses (Fig. 2d-h) because no morphological changes were found for these structures in downy mildew.

5. It is required to show morphologies of conidiospores, conidiophores, and oospores upon D-tagatose treatment, compared with the normal condition.

As described above, no morphological changes were found in conidiophores, conidiospores and oospores; they simply did not develop. Thus, we show their typical normal forms in Fig. 2g, but describe and statistically analyzed numerical data (Fig. 2d-h) because no morphological changes were found.

6. For the discussion section is too long, it is very repetitive, described too much same to the results. This section should be more focus on the importance, but not repeat the result conclusion. The authors also need to discuss about situation of similar research or application field related.

We reduced repetition with sufficient description for reader comprehension. Thank you for your suggestions.

7. Fig.S5, label problems. Fig.S6k, no error bars?

The original description was not sufficient for Fig. S5, so we rewrote the legend with revised sentences in red. We think reviewer 2 thought the description of fructokinase (LC500344) phosphorylated D-tagatose.... was shown in panel b, c of the supplementary figure, but actually the figure is Fig. 4 in the main text. We also added error bars for all data in Fig. S6. Thank you for your comments.

Reviewer #3 (Remarks to the Author):

The manuscript entitled ‘A sweet solution for an innovative, eco-friendly agrochemical using a rare sugar’ presented an interesting characterization of the mode of action of a rare sugar (tagatose) against a plant disease. The goal is interesting and experimental design is ambitious, but some aspects should be better considered in order to properly evaluate the impact and reproducibility of the obtained results.

Thank you very much for your fair criticism and encouragement on our research.

The efficacy of tagatose against plant pathogens is largely known from several years (Ohara et al. 2008 Patent JP2008209921, PCT/JP2009/003925 and Ohara et al. 2014 Patent JP2013050518. PCT/JP2014/056181). Therefore, the manuscript should be better reconsidered according to the already known information available in the literature, in order to better highlight novel results of this work. In particular,

- known properties of tagatose against plant and human pathogens should be better reported in the introduction;

Thank you for your comment. We do not know of any studies on D-tagatose and any particular effect on human pathogens. D-tagatose has been approved as safe for human consumption by multiple international organizations including WHO, FAO... etc., as we had already described. We also separated the parts with known information and highlighted more on novel results. Thank you for your comment.

- The aim of the work (lines 76-78) should better focused on novel results, i.e. the mechanism of action of tagatose;

Thank you for your comment. We revised as you suggested.

- Already known information and citations should be removed from the Results session;

We reduced repetition, but we think sufficient description is necessary for better understanding by readers, such as the difference between the effects caused by D-allose and D-allulose, which are plant activator, and those by D-tagatose, which directly affects the pathogen. For example, differences among expression patterns in plants treated with these sugars in Supplementary Fig. 3 is easier to understand by direct comparison of the patterns. Thank you for your suggestions.

- Abstract and Discussion should be dramatically tuned down and better focalized on the real results reported in this work, that are based on the mechanism of action of tagatose, but not on efficacy against plant pathogens, since efficacy is already well known and patented by the authors of this manuscript,

We reduced repetition and focused on the results and mechanism in the discussion. Thank you for your suggestions.

- Consequently, the Conclusion session (lines 337-356) appear too speculative and it is not properly

linked to the presented results and the aims declared at the beginning of this manuscript.

Thank you for your comments. Since “D-tagatose was found in the root exudates of maize seedlings, which was increased by treatment with humic acid and these exudates were predicted to influence microbial population size and community structure” is based on a published paper⁴⁵, we want to discuss the facts in relation to our finding.

Key controls seem not properly reported in the enzymatic assays. In particular:

- **profiles of pure tagatose and tagatose-6-phosphates should be reported in supplementary Figures, in order to support the key HPLC results of this manuscript;**
- **since the recombinant fructokinase is a ‘candidate’ enzyme (‘candidate’ was properly declared by the authors) the assay of its activity in the phosphorylation of fructose should be reported as control. The activity of the candidate phosphomannose isomerase in producing mannose-6-phosphate was demonstrated in Figure 3b using fructose-6-phosphate as substrate. Therefore, a similar control should be performed on the candidate fructokinase, to demonstrate that the recombinant enzyme can produce fructose-6-phosphate using fructose as substrate in vitro. Formation of fructose-6-phosphate is mentioned at lines 298-301, without visible clear indication in the Figure 4.**

Thank you very much for your comments. We fixed all the insufficiencies mentioned by reviewer 3. The particular parts revised are:

- Profiles of standard D-tagatose and D-tagatose 6-phosphate were added in Fig. 4 and aligned with HPLC profiles of respective reactions with HaFK1 to compare the retention times. All HPLC profiles of standards are given and labeled in the figure.
- We also added data for D-fructose.

The reproducibility of the data should be better clarified, in order to demonstrate robustness of the obtained results. I believe that authors appropriately carried out independent experimental repetitions with appropriate number of replicates. However, the number and type (single plant or pool of plants, single leaves or pool of leaves, tubes in case of in vitro assays) of biological replicates analyzed in each experiment should be better reported in the M&M session and in the legend of each Figure, in order to clarify how many values have been analyzed to build up the mean and standard deviation data reported in each Figure. Likewise, each experiment should be repeated at least twice in two independent repetitions and reproducibility should be declared. This information should be provided for each experiment in the M&M and in the figure legends. In particular, how experimental repetitions are used in the data analysis should be reported in the manuscript, such as statistical tests to demonstrate that experimental repetitions are comparable should be declared. Moreover, it should be better declared if one representative experiment among N. repetitions is reported in each figure or if a pool of N. different experiments is averaged and reported in each figure. This information is crucial to clarify reproducibility and possible variability of the results presented in this work. Consequently, standard

deviation values should be reported in each figure, Fig 2B, 3C, 4d, S4, S6i, S6j and S6k included. Description of number of biological replicates and independent experimental repetitions is particularly crucial for the enzymatic assays.

Thank you for comments. We added all necessary information. We originally thought the description of Fig. 2b was the easiest way for readers to comprehend the information, but we rewrote the data described in previous Fig. 2b (in Fig. 2 in initial submit) as Fig. 2c in the revised Fig 2 and included statistical evaluations.

The Materials and Methods section is unclear and descriptions should be improved in order to easily allow the repetition of the experiments by other groups.

- Some key methods are reported in the supplementary materials (enzymatic assays), while other secondary methods (efficacy of tagatose against pathogens) are reported in a confused manner in the main document.

- If tagatose efficacy against pathogens is included in the manuscript, methods should be described appropriately.

Descriptions are scarce and insufficient to repeat the experiment. moreover, some key protocols cited in the manuscript are not accessible, such as conference proceedings of citation 50, or not easily understandable form an international scientific community, such as the manual written in Japanese of citation 51. Please consider that information to easily repeat each experiment for each assay against each pathogen should be provided or international scientific community. Therefore, descriptions should be improved and accessible citations should be mentioned.

Thank you for these comments. We revised all points suggested and clarified our points. Assay methods are now detailed so that they can be repeated by readers. Because we intend to develop a commercial D-tagatose product in Japan, we need to complete our assay according to the suggested methods in the reference by Japanese regulatory agency; the methodology is basically the same as that required by agencies in many countries, including e.g., EPPO. Again we described the methods for the assays in detail in the manuscript and Supplementary Tables. We think every country has a similar manual but not all are translated in English, including the Japanese, and we have no official English translation. Thank you very much for your understanding.

- Protocols used to stain *H. arabidopsidis* in Arabidopsis leaves and to assess the disease severity are not appropriately described in the M&M. Moreover, the +, ++ and +++ classes should be converted to a more appropriate, and possibly objective, symptom class description.- Likewise, description of direct effect assay on *H. arabidopsidis* (lines 462-471) is not clear and not supported by appropriate citations.

Thank you for your comment. Hyphal growth of *H. arabidopsidis* in Arabidopsis leaves was assessed as the extent of colonization in leaves as a percentage of leaf area as follows, and the method is now detailed in the methods section, results and figure legend. We also described our unique method for germination using Gamborg B5 for *H. arabidopsidis*.

- Gene expression description is a mixture of different protocols, with missing parts on statistical analysis. On the other hand, some other parts include an excess in detail even though they are really standard protocols, such as the library preparation of Illumina which is well known. Please consider to avoid description of really standard protocols but improve descriptions of specific protocols used in this work, particularly on statistical analyses. For example, the statistical analysis to select the 288 clone upregulated in tagatose-treated leaves is missing and it should be clearly reported.

Thank you for the comment. We reduced some protocol descriptions and described others in more detail as suggested. We also described all necessary statistical analyses including, e.g., Figs. 2c, 3b, 3c (combined data are shown in Fig. 3 c and Supplementary Fig. 4a), and also SD additions for Fig. 4d and Supplementary Fig. 6.

A random selection is indicated, but cut off used to identify significant up-regulations should be provided. Likewise, statistical analysis used to compare mock-treatment and sugar-treatment on microarray results is missing.

Thank you for this comment. We changed the figure description and added results of statistical analyses ($n = 3$) of Tukey–Kramer’s test using JMP12 software.

Once again, statistical analysis on RNA-Seq data is missing. DeSeq or similar statistical softwares are commonly used to detect significant up- and down-regulated genes in RNA-Seq analysis. Therefore, the number of replicates and statistical tests of RNA-Seq data should be reported in this manuscript.

Thank you. As previously described in the materials and methods, we isolated RNA from 20 to 30 independent leaves as biological repeats of respective experiments, and Cuffdiff (v2.0.2) was used to analyze each fold change and q -value (the FDR-adjusted uncorrected p -value of the test statistic used to compute significance of the observed change in FPKM) obtained from these numbers of fragments. The statistical data are summarized and added in Supplementary Fig. 4, and a simplified color-imaged description is shown in Fig. 3c. We submitted total RNA-seq data of respective analyses to a databank for public access as described in the text.

Unclear descriptions can be found in the phytopathological part and this dramatically preclude to understand the confidence of the presented data. *H. arabidopsidis* is a well-known obligate biotroph, which growth only in presence of the alive host. Therefore, the in vitro protocol to germinate conidiophores in Gamborg B5 medium (lines 462-471 and 588-591) should be supported by appropriate references. In other words, no germination of an obligate biotroph is expected in vitro; the protocol of *H. arabidopsidis* in vitro seems not properly performed/described. Therefore, since *H. arabidopsidis* cannot growth in vitro, the cell amount used for the crude extract assay is probably really low.

Thank you for your comments. As we wrote above, hyphal growth has been examined even for obligate biotrophs including for example *Peronospora parasitica* (Achhar, 1997, J. Phytopathol. 146, 137-141), *Peronospora belbahrii* (Cohen and Ben-Naim, 2016, PLoS ONE, 11(5), e0155330), *Peronospora tabacina*

(Shepherd, 1962, Aust. J. Biol. Sci., 15, 483-510), and *Puccinia recondita* (our lab.), *Puccinia coronata* (personal communication) are also known to germinate and hyphal growth can be observed in water, although of course, none of biotrophs can be maintained on the medium. No appropriate methodology exists for *H. arabidopsidis*; we thus described a method to germinate conidiospores using Gamborg B5.

An estimation of the real quantity of tagatose-6-phosphate in the crude extract, expressed as micro gram per mg of fresh weight (or per mg of total proteins) is expected in order to clarify the quantification of Figure 4a.

Thank you for your comment. All fungal samples started from 2×10^6 spores were used to prepare crude enzymes as described in Methods. The results are solid and detectable but not appropriate for relative comparison. Thus, we isolated the enzyme by cloning the gene encoding the target enzyme for further analyses.

Minor comments:

- Introduction describes several sugars and rare sugars, while it should be better focalized on tagatose properties and known information.

Thank you for the comment. We think even the term of rare sugar is still not generally familiar to most readers. We kept a few sentences on several sugars and rare sugars and the use of rare sugars for plant protection as necessary to explain why we started to use D-tagatose, but minimized the descriptions as suggested.

- Figures and Tables as usually cited in the Results, The used of Figure and Table citations in Introduction and discussion is discourage, to avoid confusions on novel data and data already reported in the bibliography.

Thank you for your comments. We added a few Supplementary Figs. in the section for the benefit of readers, but we think there is little chance of confusion regarding the structures of these sugar and new concepts that are probably not yet common knowledge, and thus we chose to keep them.

- On the other hand, citations and introduction like sentences (e.g. lines 121-128, 136-144, 170-179) should be avoided in the Results session.

Thank you for your comments. We think it helps to understand the reason or concept for experiments, but we shortened these parts as suggested.

- Some results are described in the M&M and figures are cited as well, and this dramatically reduces the readability of this session.

Thank you for your comments. The figures cited in the M&M are sugar structures, a list of pathogens used, and explanations for figures that can be confused with each other, but we shortened this section.

- The M&M on enzymes characterization (lines 586-636) is a mixture of cDNA cloning and assays, I suggest to split it in different paragraphs to distinguish cloning session, assays on crude extracts and assays on purified recombinant proteins.

Thank you for your comments. We split them as suggested into two parts: "cDNA cloning of enzyme genes" and "Characterization of enzymes for monosaccharide phosphorylation and inhibition of activity by D-tagatose" section.

- Table 1 contains a mixture of published results and novel results, I suggest to mention published results in the introduction and Discussion, and to clarify better the novel results provided in this work.

Thank you for your comments. We have added references for our previous work and other papers, and we think there is little chance for confusion with our new findings.

- Four conditions are mentioned for Figure 3C (control, tagatose treatment and inoculation with the pathogen on both treatments), but only a single mean value is reported for each gene. All fold change values (with standard deviation) should be reported for each pairwise comparison, and significant changes should be highlighted.

Thank you for your comments. This experiment is designed as a comparison of two conditions on inoculated leaves: with (one)/or without (another) D-tagatose treatment. Changes in total expression patterns in rice plants after D-tagatose treatment were analyzed using microarray (this work), and there are many published data for the expression patterns of Arabidopsis with or without inoculation of this pathogen (e.g., Hok *et al.*, 2011, Plant Cell Environ., 34, 1944-1957; Asai *et al.*, 2014, PLoS Pathog., 10 (10), e1004443). As we described above, we isolated RNA from 20 to 30 independent leaves as biological repeats of respective experiments, and used Cuffdiff (v2.0.2) to analyze each fold change and q -value (the FDR-adjusted uncorrected p -value of the test statistic was used to compute significance of the observed change in FPKM) obtained from these numbers of fragments. They are summarized in Supplementary Fig. 4 and described with a simplified color image in Fig. 3c. We submitted total RNA-seq data of respective analyses in a databank with public access as described in the text. No significant differences (significance level of 0.05) were found except for *At2g14247*, q -value = 0.0021 (Supplementary Fig. 4b) with annotation of an unknown expressed protein, and none of the expression patterns of genes encoding those with various known functions changed significantly (Supplementary Fig. 4a).

- Statistical analysis is missing in Figure 3b, 3c and 4d.

Thank you for your comments. We added all necessary statistics.

- Results description of lines 12-147 should be improved, in order to better discriminate already published results from novel data. The use of citations and published results is preferred for the Discussion instead of Results session.

Thank you for your comments. As we stated above, we think it helps to understand the reason or concept for the experiments, but we shortened the sentences as suggested.

- Conclusion at lines 272-275 is not supported neither by appropriated references nor results of this manuscript.

Thank you for your comments. Although we think our present results support the statement, we omitted these sentences because they are not the main topic of this paper.

REVIEWERS' COMMENTS:

Reviewer #2 (Remarks to the Author):

I think the authors have addressed all of my concerns.

Reviewer #3 (Remarks to the Author):

The revised manuscript entitled 'A sweet solution for an innovative, eco-friendly agrochemical using a rare sugar' presents important changes that dramatically increased readability and impact of the presented results. As clarification of some of my previous comments here below you can find some minor suggestions.

As previously suggested, already known properties of tagatose against plant and human pathogens should be better reported in the introduction and discussion session. This could also help to better describe the possible mode of action against plant pathogens, taking information from knowledge derived from other systems. For example, the well-known inhibitory effect of tagatose on human pathogens (Hasibul et al. 2018 doi: 10.3892/mmr.20178017, Bautista et al. 2000 doi: 10.4315/0362-028X-63.1.71; Lobete et al. 2017 doi: 10.1016/j.foodcont.2016.05.049), the mechanism of action of tagatose on human-associated microorganisms (Koh et al. 2013 doi: 10.1016/j.fm.2013.03.003; Martinussen et al., 2013 doi: 10.1016/j.copbio.2012.11.009; Wu et al. 2017 doi: 10.1016/j.fm.2016.10.027) and the stimulation of beneficial human associated microorganisms (Hasibul et al. 2018 doi: 10.3892/mmr.20178017; Vastenavond et al. 2011 doi: 10.1201/b11242-15) should be better considered in the introduction and in the discussion of the possible modes of action of tagatose. Likewise, some results on tagatose effects against a grapevine oomycete pathogen (Perazzolli et al. 2020 doi: 10.1016/j.micres.2019.126387) and a tomato oomycete pathogen should be better considered (Chahed et al. 2020 doi: 10.3389/fmicb.2020.00128) in this manuscript. For example, possible combined mechanisms of action of tagatose on mitochondrial respiration (Chahed et al. 2020 doi: 10.3389/fmicb.2020.00128) and mannose metabolism (this submitted work) of plant-associated oomycetes should be better discussed in this manuscript.

Sentence at line 197-198 is not appropriated, since pathways of tagatose metabolism have been really well documented in other systems (e.g. Van Der Heiden et al. 2013 doi: 10.1128/AEM.03918-12, Wu et al. 2017 doi: 10.1016/j.fm.2016.10.027). An appropriated discussion of tagatose metabolism is suggested.

The other major comments previously suggested have been fully addressed, thanks to the proper revision of the manuscript carried out by the authors.

Dear reviewer#3 of Communications Biology,

Thank you very much for your favorable review and helpful criticism of our manuscript. The manuscript is now much better after the revision. Our responses to the advice and questions follows.

Comments from Reviewer#3

As previously suggested, already known properties of tagatose against plant and human pathogens should be better reported in the introduction and discussion session. This could also helpful to better describe the possible mode of action against plant pathogens, taking information from knowledge derived from other systems. For example, the well-known inhibitory effect of tagatose on human pathogens (Hasibul et al. 2018 doi: 10.3892/mmr.20178017, Bautista et al. 2000 doi: 10.4315/0362-028X-63.1.71; Lobete et al. 2017 doi: 10.1016/j.foodcont.2016.05.049), the mechanism of action of tagatose on human-associated microorganisms (Koh et al. 2013 doi: 10.1016/j.fm.2013.03.003; Martinussen et al., 2013 doi: 10.1016/j.copbio.2012.11.009; Wu et al. 2017 doi: 10.1016/j.fm.2016.10.027) and the stimulation of beneficial human associated microorganisms (Hasibul et al. 2018 doi: 10.3892/mmr.20178017; Vastenavond et al. 2011 doi: 10.1201/b11242-15) should be better considered in the introduction and in the discussion of the possible modes of action of tagatose. Likewise, some results on tagatose effects against a grapevine oomycete pathogen (Perazzolli et al. 2020 doi: 10.1016/j.micres.2019.126387) and a tomato oomycete pathogen should better considered (Chahed et al. 2020 doi: 10.3389/fmicb.2020.00128) in this manuscript. For example, possible combined mechanisms of action of tagatose on mitochondrial respiration (Chahed et al. 2020 doi: 10.3389/fmicb.2020.00128) and mannose metabolism (this submitted work) of plant-associated oomycetes should be better discussed in this manuscript.

Thank you very much for your various comments. We appreciate it very much, and our manuscript is much better after taking your suggestions. We revised according to your suggestion with the following comments in last review “- **Abstract and Discussion should be dramatically tuned down and better focalized on the real results reported in this work, that are based on the mechanism of action of tagatose, but not on efficacy against plant pathogens, since efficacy is already well known and patented by the authors of this manuscript**”, and we reduced significantly the contents which are not directly related to our findings. Also explanations of other general well-known matters were minimized according to your suggestion, and we tried to use review-like references for respective descriptions. References such as 30 – 40 cover all topics necessary to discuss contents of our findings in our model. Since we agreed to you by taking the suggestion in your last review and the contents were simplified to read easier for readers now, we appreciate your suggestion but we would like to avoid re-addition of other topics for D-tagatose or the effects which are not directly related to our findings.

Sentence at line 197-198 is not appropriated, since pathways of tagatose metabolism have been really well documented in other systems (e.g. Van Der Heiden et al. 2013 10.1128/AEM.03918-12, Wu et al. 2017 doi: 10.1016/j.fm.2016.10.027). An appropriated discussion of tagatose metabolism is suggested

Thank you for the comment. As reviewer knows, D-tagatose is one of intermediate substances in lactose (galactose) metabolism and the metabolic pathway based on the flow from lactose to F6P in milk is described frequently as tagatose metabolism including the references suggested by the reviewer. However, D-tagatose itself rarely exists alone in nature as the definition of rare sugars, in which rare sugars are monosaccharides that are rarely present in nature. Thus, no enzyme or enzyme-encoding gene specific for rare sugar as the only substrate has been identified, and the enzymes generally have multiple sugar substrates including a target rare sugar like D-tagatose. Our strategy to identify the enzymes which can take D-tagatose as their one of substrates is reasonable, and we would like to give no change on this sentence.